# Targeting FAT1 Inhibits Carcinogenesis, Induces Oxidative Stress and Enhances Cisplatin Sensitivity through Deregulation of LRP5/WNT2/GSS Signaling Axis in Oral Squamous Cell Carcinoma

**DOI:** 10.3390/cancers11121883

**Published:** 2019-11-27

**Authors:** Tung-Nien Hsu, Chih-Ming Huang, Chin-Sheng Huang, Mao-Suan Huang, Chi-Tai Yeh, Tsu-Yi Chao, Oluwaseun Adebayo Bamodu

**Affiliations:** 1Division of Oral and Maxillofacial Surgery, Department of Dentistry, Taipei Medical University―Shuang Ho Hospital, New Taipei City 235, Taiwan; 2School of Dentistry, College of Oral Medicine, Taipei Medical University, Taipei City 110, Taiwan; 3Department of Otolaryngology, Taitung Mackay Memorial Hospital, Taitung City 950, Taiwan; 4Department of Hematology and Oncology, Cancer Center, Taipei Medical University—Shuang Ho Hospital, New Taipei City 235, Taiwan; 5Department of Medical Research & Education, Taipei Medical University―Shuang Ho Hospital, New Taipei City 235, Taiwan; 6Graduate Institute of Clinical Medicine, School of Medicine, Taipei Medical University, Taipei City 110, Taiwan; 7Taipei Cancer Center, Taipei Medical University, Taipei City 110, Taiwan

**Keywords:** OSCC, squamous cell carcinoma, FAT1, atypical cadherin, oncogene, chemoresistance, cisplatin, Wnt signaling, LRP5, oxidative stress, GSH, GSS

## Abstract

FAT atypical cadherin 1 (FAT1) regulates cell-cell adhesion and extracellular matrix architecture, while acting as tumor suppressor or oncogene, context-dependently. Despite implication of FAT1 in several malignancies, its role in oral squamous cell carcinoma (OSCC) remains unclear. Herein, we document the driver-oncogene role of FAT1, and its mediation of cell-death evasion, proliferation, oncogenicity, and chemoresistance in OSCC. In-silica analyses indicate FAT1 mutations are frequent and drive head-neck SCC, with enhanced expression defining high-risk population and poor prognosis. We demonstrated aberrant FAT1 mRNA and protein expression in OSCC compared with non-cancer tissues, whereas loss-of-FAT1-function attenuates human primary SAS and metastatic HSC-3 OSCC cell viability, without affecting normal primary human gingival fibroblast cells. shFAT1 suppressed PCNA and upregulated BAX/BCL2 ratio in SAS and HSC-3 cells. Moreover, compared with wild-type cells, shFAT1 concomitantly impaired HSC-3 cell migration, invasion, and clonogenicity. Interestingly, while over-expressed FAT1 characterized cisplatin-resistance (CispR), shFAT1 synchronously re-sensitized CispR cells to cisplatin, enhanced glutathione (GSH)/GSH synthetase (GSS)-mediated oxidative stress and deregulated LRP5/WNT2 signaling. Concisely, FAT1 is an actionable driver-oncogene in OSCC and targeting FAT1 in patients with erstwhile cisplatin-resistant OSCC is therapeutically promising.

## 1. Introduction

Oral cancer, comprising cancers of the oral cavity (International Statistical Classification of Diseases and Related Health Problems, ICD-10: C00-06) and oropharyngeal regions (ICD-10: C09-10), is one of the most prevalent malignancies in the world, with an annual incidence of 447, 751 cases [1], and with a predicted 67.1% rise in the disease-specific mortality by 2035 (*n* = 242, 886) compared with 2012 (*n* = 145, 353), oral cancer remains a principal cause of cancer-related mortality globally [1,2]. Though multifactorial, the commonest risk factors associated with oral cancer include gene susceptibility, betel nut chewing, tobacco smoking, alcohol abuse, and human papillomavirus (HPV) infection [3,4]. The oral squamous cell carcinoma (OSCC) histological type constitutes more than 90% of all oral cancer, and is highly invasive, most often insensitive to chemo- and/or radiation therapy, and associated with high incidence of recurrence, and poor survival rates [1,2].

Currently, the standard of care remains surgery for patients with early (I and II)―or advanced (III and IV)―stage tumors, and definitive chemoradiotherapy for patients with advanced-stage tumors, where chemotherapy consists of cisplatin (CDDP), docetaxel, and 5-flurouracil, especially for patients with advanced-stage malignancies [5,6], however, this is often associated with increased risk of severe therapy-related toxicities and adverse effects, including neutropenia and osteoradionecrosis, increasing incidence of therapy failure and disease relapse, and low median survival rates for patients with OSCC [3], as indicated by a 5-year survival rate that has remained consistently below 50% over the last 3 decades [7].

Despite the association of adjuvant chemotherapy with enhanced survival of patients with advanced stage OSCC, and the touted therapeutic promise of cisplatin (*cis*-diamminedichloroplatinum, CDDP), an alkylating chemotherapeutic agent, with well-documented anticancer efficacy for patients with cancers, including OSCC, intrinsic insensitivity or acquired resistance to cisplatin and disease recurrence continues to impede treatment success and plague survival of patients with OSCC [8]. Thus, the need to unravel probable mechanisms underlying cisplatin-resistance, which invariably will enable the discovery of novel actionable molecular or therapeutic targets and development of more effective treatment strategies to abrogate chemoresistance and improve the clinical outcome of patients with OSCC. Recent work by Mascitti et al, very well encapsulates the epidemiological implication and prognostic ramifications of this dearth of therapeutically-relevant actionable targets and prognostigators, and further begs the case for same [9].

The last two decades has been characterized by increased interest in and accumulating evidence of the critical role of the tumor microenviroment (TME) in the initiation and progression of cancer, as well as the response of cancerous cells to therapeutic modality [10,11,12]. Cell adhesion molecules, including cadherins are vital components of the TME, being associated with cell-cell adhesive bonds in solid tissues, immune modulation, promotion of cancer growth, metastasis and survival [13] The role of atypical cadherins such as the FATs in tumor formation, dissemination, and prognosis, has garnered attention lately. The human *FAT* gene family, consisting of *FAT1*, *FAT2*, *FAT3* and *FAT4* genes, encodes large proteins with extracellular Cadherin repeats, EGF-like domains, Laminin-G-like domains, where human FAT1, FAT2, and FAT3 are orthologous with *Drosophila* Fatl, and human FAT4 arthologous with *Drosophila* Fat [14]. While there is ample information on the role(s) of the FAT atypical cadherin 1 (FAT1) in contemporary literature, these information are rather conflicting, with FAT1 suggested as a tumor suppressor based on its inhibition of Yes-associated protein (YAP)1 function and suppression of cell growth [15], inhibition of epithelial-to-mesenchymal transition (EMT) in esophageal squamous cell cancer [16], suppression of the invasive capability, nodal involvement, lymphovascular permeation and tumor recurrence in HNSCC [17], and its loss-of-function eliciting resistance to cyclin dependent kinase (CDK)4/6 inhibitors in ER+ breast cancer [18]. Conversely, aberrant expression of FAT1 has been implicated in the high invasiveness of GBM cells [19], cancerous cell proliferation, apoptosis evasion and disease progression in hepatocellular carcinoma (HCC) [20], relapse and poor prognosis in patients with B-cell acute lymphoblastic leukemia [21].

Against the background of this ambivalent context-dependent role of FAT1 in malignancies and its under-explored role in OSCC, the present study investigated the probable implication of FAT1 in the oncogenicity, metastatic and therapy-resistance phenotypes of OSCC cells and the poor prognosis of patients with OSCC. Herein, we demonstrated that aberrantly expressed FAT1 by cancerous cells enhanced their proliferation, promoted chemoresistance, impaired cisplatin-induced cell death, and that the therapeutic targeting of FAT1 re-sensitized cisplatin-resistant OSCC cells to cisplatin through deregulation of LRP/Wnt signaling, thus projecting FAT1 as a novel therapeutic target for anticancer treatment of therapy-resistant OSCC.

## 2. Results

### 2.1. High FAT1 Expression Drives HNSC, Defines High Risk Population and is Associated with Poor Prognosis

In the light of the divergent roles of FAT1 in different malignancies, seeking to understand the pathocytological relevance of FAT1 and determine its molecular dynamics in highly metastatic and recurrent OSCC cells, we examined FAT1 expression and mutational profile in the TCGA HNSC cohort (*n* = 502). Results of our bioinformatics analysis showed that of the 131 mutated/mutant cancer drivers detected in the TCGA HNSC cohort, the most mutated drivers included TP53, FAT1, NOTCH1, CDKN1, CDKN2A and PIK3CA, in decreasing order of mutational frequency (Figure 1A). In addition, we observed that of the principal 97 mutations in FAT1, the truncating (otherwise known as nonsense) gene mutation accounting for 75% of all FAT1-associated mutations, was most frequent (Figure 1B). Because of the implication of driver mutation burden in poor clinical outcome [22], we evaluated the effect of altered FAT1 expression in the TCGA HNSC cohort, and demonstrated that compared to the FAT1^low^ group, high FAT1 (FAT1^high^) expression confers significant survival disadvantage on the patients with HNSC ((hazard ratio (HR) = 1.34, 95% CI: 1.02–1.77; *p* = 0.038) (Figure 1C). Consistent with the above, the FAT1^high^ group exhibited strong association with high risk of disease-specific mortality compared with the FAT1^low^ group (Figure 1D), suggesting a prognostic role for FAT1 in patients with HNSC.

### 2.2. High FAT1 Expression Is Positively Correlated with Disease Progression and Poor Clinical Outcome in Patients with HNSC

In the light of our data suggesting a driver and prognostic role for FAT1 mutations and expression, respectively, in patients with HNSC, probing further for validation of these function, we observed a strong association between increased expression of FAT1 (probe ID: 201579_at) and disease progression, as demonstrated by mean expression transformed counts of 2.31, 1.07, 1.2, and 1.46 in stage I (T1), stage II (T2), stage III (T3) and stage IV (T4), compared with 0.97 in normal samples (Figure 2A), based on analysis of the GSE3524/GDS1584 dataset of gene expression profiles in OSCC cells from 16 patients with 20 samples, isolated by laser capture microdissection. Interestingly, FAT1 expression spikes in this dataset were markedly associated with nodal involvement as seen in T1N2bM0, T3N2bM0, T4N1M0, and T4N2bM0 (Figure 2A, also see Appendix A). Our analysis of the OSCC cohort (*n* = 291) of the TCGA HNSC dataset showed that aberrant expression of FAT1 in patients with OSCC was strongly associated with worse overall survival in the FAT1^high^, compared to the FAT1^low^ patients [HR: 1.12 (0.97–1.3), *p* = 0.121] with the median survival for FAT1^high^ and FAT1^low^ patients being 1081 and 1591 days, respectively (Figure 2B). Moreover, analysis of the E-MTAB-1328 dataset from methylome, transcriptome and miRNome array and high throughput sequencing profiling of 89 patients with 104 samples also showed that FAT1^high^ patients exhibited shorter metastasis-free survival than their FAT1^low^ counterparts [HR:1.1 (0.74–1.63), *p* = 0.653] (Figure 2C) and shorter relapse-free survival [HR:2.35 (0.48–11.52), *p* = 0.290] in the FAT1^high^, compared to the FAT1^low^ patients, based on analysis of 44 primary head and neck tumor samples using the GSE10300 dataset (Figure 2D). These data are indicative of a critical role of high FAT1 expression in disease progression and poor clinical outcome in patients with HNSC.

### 2.3. FAT1 Is Aberrantly-Expressed in OSCC Clinical Samples and Cell Lines

Since the focus of the present study is more specifically OSCC, tapering down from our evolving generic understanding of the role of FAT1 in HNSC, using the Peng Head-Neck OSCC cohort (*n* = 79), we further demonstrated a 2.82-fold increase in FAT1 expression in the OSCC compared to the ‘normal’ non-tumor oral cavity samples (*t*-test = 12.0, *p* = 4.57 × 10^–18^) (Figure 3A). Similarly, analysis of the Estilo Head-Neck tongue squamous cell carcinoma (TSCC) cohort data (*n* = 58) showed a 2.99-fold increase in the expression of FAT1 in TSCC samples compared to normal tongue samples (*t*-test = 5.384, *p* = 8.77 × 10^−7^) (Figure 3B). Immunohistochemical (IHC) staining of samples from our OSCC tissue archive demonstrates significant overexpression of FAT1 protein in the plasma membrane and nucleus of cancerous cells, compared to the neighboring non-cancerous cells (Figure 3C). Consistent with our IHC data, results of our western blot analyses of our OSCC samples indicate the FAT1 protein is preferentially highly expressed in the cancerous cells compared to their non-cancerous counterparts (Figure 3D). On the transcript level, we also demonstrate a 2.04-fold enhancement in the mRNA expression of FAT1 in our OSCC samples compared with the non-cancerous samples (*p* < 0.01) (Figure 3E). In addition, using *in vitro* OSCC models, we demonstrated that compared to its expression in adult human primary normal gingival fibroblast cells (HGF, ATCC® PCS-201-018™), the expression of FAT1 protein was significantly up-regulated in the highly metastatic human OSCC HSC-3 (2.14-fold, *p* < 0.05), and SAS (2.76-fold, *p* < 0.01) cells (Figure 3F). These data do indicate that FAT1 is aberrantly expressed in OSCC clinical samples and cell lines.

### 2.4. Loss-of-FAT1 Function Impairs OSCC Cell Proliferation and Enhances Cell Death

To proffer a solution, we investigated if and to what degree the loss-of-FAT1 function negatively affects the survival and proliferation of OSCC cells. Our results indicate that shFAT1 transfection markedly suppressed the proliferation *cum* viability of SAS (62.5%, *p* < 0.05) or HSC-3 (58.1%, *p* < 0.05) cells, compared to their wild-type counterparts; interestingly the repressive effect of shFAT1 on the proliferation/viability of normal gingival fibroblast cell line HGF, was barely apparent and statistically insignificant (Figure 4A), indicating the non-lethality of shFAT1 to normal oral cavity cells, and at least in part, the oncoselectivity of the loss-of-FAT1 function in patients with OSCC. Using Ki-67 protein immunofluorescence staining, we demonstrated further that shFAT1 induced 80.6% (*p* < 0.01) and 78.3% (*p* < 0.01) reduction in the number of viable SAS and HSC-3 cells, respective, while conversely having no effect on the HGF cells which increased by 6% (Figure 4B). In parallel assays, against the background of cell-cycle regulated and apoptosis stimuli-mediated existent interconnection between cell proliferation and apoptosis [23], we demonstrated that shFAT1 significantly down-regulated the expression levels of marker of proliferation Ki-67, the proliferating cell nuclear antigen (PCNA) and B cell lymphoma 2 (Bcl2) proteins, while up-regulating Bax protein expression in the transfected SAS and HSC-3 cells, compared to their wild-type counterparts (Figure 4C), which is suggestive of a role for FAT1 in the promoting OSCC cell proliferation and evasion of cell death.

Furthermore, gene expression-based heatmap generated from re-analysis of the A-AFFY-44, AFFY_HG_U133_PLUS_2, E-GEOD-30784 dataset of the gene expression profile of OSCC cohort (*n* = 229 samples, 54675 genes), originally to identify potential biomarkers for early detection of invasive OSCC using OSCC (*n* = 167), oral dysplasia (*n* = 17) and normal oral (*n* = 45) samples, demonstrated that FAT1 expression was marginal in the non-cancerous control samples, but high FAT1 expression was associated with high PCNA, BIRC5/Survivin, BCL2, and BCL2L1, while low FAT1 expression correlated with high FADD, BAX, PARP1, CASP3, CASP8, and CASP9 in the cancer tissues (Figure 4D) suggesting an association between reduced FAT1 expression and induction of cell death. For a balanced perspective, we also observed that BAX was high in some of the FAT1^high^ samples, however the median BAX/BCL2 ratio in the FAT1^high^ samples was ~2.83-fold higher than in the FAT1^low^ samples (Figure 4D). Singular value decomposition (SVD)-calculated principal component analysis (PCA) of same E-GEOD-30784 OSCC cohort further shows that while most (95.8%) OSCC sample were FAT1^high^, 4 were FAT1^low^ (2.4%), and 3 were ambiguous (Figure 4E). These results indicate that the loss-of-FAT1 function significantly impairs the proliferation of OSCC cells and enhances apoptosis, but spares the normal or non-malignant cells of the oral cavity.

### 2.5. Loss-of-FAT1 Function Attenuates the Oncogenicity and Metastatic Phenotype of OSCC Cells

Having shown a positive correlation between high FAT1 expression and HNSC, as well as implicated the aberrant expression of FAT1 in HNSC progression and poor clinical outcome in patients, we sort to gain some insight into its mechanistic underlining. Our assessment of FAT1 oncogenic activity using established functional assays revealed that shRNA-mediated silencing of FAT1 (shFAT1) significantly suppressed the motility of HSC-3 (2.86-fold, *p* < 0.05) and SAS (2.69-fold, *p* < 0.05) cells, compared to the wild-type control cells after 24 h (Figure 5A). Similarly, shFAT1 elicited a 2.97-fold (*p* < 0.05) and 7.36-fold (*p* < 0.01) attenuation of invasion in SAS and HSC-3 cells, respectively, compared to their wild-type counterparts (Figure 5B). Moreover, we also demonstrated that compared to the wild-type control cells, shFAT1 markedly inhibit the clonogenicity of SAS (8.5-fold, *p* < 0.01) and HSC-3 (9.33-fold, *p* < 0.01) cells (Figure 5C). Consistent with these data, our re-analysis of the Holsinger GPL570/GSE42743/GSM1049079 Oral Cavity Cancer cohort (*n* = 103) using the we further demonstrated moderate to strong positive correlation between the expression of FAT1 and N-cadherin/CDH2 (*R* = 0.17, *p* = 0.09), β-catenin/CTNNB1 (*R* = 0.24, *p* = 0.01), slug/SNAI2 (*R* = 0.68, *p* = 1.92 × 10^-15^), vimentin/VIM (*R* = 0.23, *p* = 0.02), or lysine-specific demethylase 5B/KDM5B (*R* = 0.40, *p* = 3.32 × 10^-5^), while being inversely correlated with E-cadherin/CDH1 (*R* = 0.06, *p* = 0.57) (Figure 5D). Interestingly, using the STRING version 11.0 (https://string-db.org), our protein-protein network visualization of probable molecular interaction between FAT1, CDH1, CDH2, CTNNB1, SNAI2, VIM, and KDM5B revealed an average node degree of 15.1°, average local clustering coefficient of 74.6%, and protein-protein interaction (PPI) enrichment *p*-value of < 1.0 × 10^−16^ (Appendix A), indicating that the proteins have more interactions among themselves than would be expected in a random set of proteins with similar size, drawn from the genome. Thus, indicating that the proteins are at least partially biologically connected, as a group. These data, at least in part, indicate that FAT1 via interaction with and/or regulation of CDH1, CDH2, CTNNB1, SNAI2, VIM, and KDM5B, modulate oncogenicity and metastatic phenotype of OSCC cells.

### 2.6. Molecular Targeting of FAT1 Re-Sensitizes Cisplatin-Resistant OSCC Cells to Cisplatin through Enhanced Oxidative Stress and Deregulation of LRP/Wnt Signaling

Having demonstrated the implication of enhanced FAT1 expression in OSCC cell proliferation, oncogenicity, metastatic phenotype, and evasion of cell death, we sought to understand if, how and to what extent FAT1 expression is implicated in resistance to cisplatin treatment using adaptive cisplatin resistant (CispR) OSCC cells.

Our results indicated that compared to the wild-type (WT) cells, the HSC-3 CispR and SAS CispR were significantly less responsive to cisplatin treatment, with concentration as high as 40 μM eliciting only a ~38.9% (vs. 68.1% in WT, *p* < 0.05) or 41.5% (vs. 87.4% in WT, *p* < 0.01) reduction in the viability of the HSC-3 CispR and SAS CispR, respectively (Figure 6A). Thereafter, we demonstrated that FAT1 is implicated in this cisplatin resistance phenotype with a 7.4-fold higher mRNA expression in the HSC-3 CispR cells compared to the HSC-3 WT cells (*p* < 0.01) (Figure 6B), and a 3.7-fold (*p* < 0.01) or 4.8-fold (*p* < 0.01) upregulated FAT1 protein expression in the SAS CispR or HSC-3 CispR cells compared to their parental/wild-type counterparts (Figure 6C). Consistent with the data above, and in line with contemporary knowledge implicating increased glutathione (GSH) levels in tumor initiation, disease progression, increased metastasis, and the chemoresistant stem cell-like phenotype of cancerous cells [24,25,26], we showed that the median intracellular GSH level in the FAT1-rich HSC-3 CispR cells was 2.38-fold (*p* < 0.01) fold higher than in the HSC-3 wild-type cells (Figure 6D). Conversely, shFAT1 induced ~ 60% reduction in the antioxidant GSH protein expression of HSC-3 CispR cells, compared to their wild-type control counterparts (Figure 6E), and was associated with significant enhancement of cisplatin-induced inhibition of OSCC cell viability, as demonstrated by a 50%, ~80%, and ~96% loss of viability of the HSC-3 CispR shFAT1 cells at concentrations of 10 μM, 25 μM, and 40 μM, respectively (Figure 6F). Moreover, in a bid to gain some mechanistic insight into the anticancer activity of loss of FAT1 function, we probed molecular components of our predicted FAT1 interactome (Appendix A), our results showed that while 5 μM cisplatin downregulated the expression of low-density lipoprotein (LRP)5, p-GSK3β, GSK3β and active β-catenin proteins by 26% (*p* < 0.05), 31% (*p* < 0.05), 5% (*p* = 0.869), and 47% (*p* < 0.01), respectively, combining shFAT1 with 5 μM cisplatin significantly enhanced the protein expression-suppressing effect of the later, as shown by 67% (*p* < 0.01), 72% (*p* < 0.001), 88% (*p* < 0.001), and 81% (*p* < 0.001) decrease in LRP5, *p*-GSK3β, GSK3β and active β-catenin proteins’ expression levels, with a predisposition for dose-dependence (Figure 6G); which is corroborated by findings from our bioinformatics analyses showing strong co-expression of FAT1, molecular components of the LRP5 signalosome, namely LRP8, GSK3β, WNT2, β-catenin, casein kinase 1 gamma 1 (CSNK1G1), CSNK1G2, CSNK1G3, AXIN1, caveolin (CAV)1, CAV2, and glutathione (GSS), with weak expression of glutathione-disulfide reductase (GSR) in patients with OSCC, compared with the dysplasia or normal control group (Appendix A), indicating, at least in part, that the molecular targeting of FAT1 re-sensitizes cisplatin-resistant OSCC cells to cisplatin treatment by deregulating the LRP/WNT signaling pathway. Interestingly, we also observed a strong correlation between the FAT1^high^LRP5^high^CTNNB1^high^GSS^high^ phenotype and poor clinical outcome, as shown by ~32%, ~64%, and 100% survival disadvantage at the 50th, 100th and 150th months, respectively, compared to their FAT1^low^LRP5^low^CTNNB1^low^GSS^low^ counterparts [concordance index = 64.15, hazard ratio with confidence interval of 2.75 (1.28–5.92), *p* = 0.009] (Appendix A). This data is consistent with the Schrödinger PyMOL 2.3 (https://pymol.org/2/) generated direct interaction between FAT1, LRP5 and glutathione synthetase (GSS, GSH-S) to form the FAT1/LRP5/GSS signalosome complex with a complex-formation or docking score of −241.20, and ligand root-mean-square deviation of 58.44Å in OSCC cells (Appendix A).

## 3. Discussion

OSCC remains a principal cause of morbidity and mortality in patients with HNSC, especially considering that in spite of documented decrease in OSCC incidence over the last decade, improvement in overall survival as a primary clinical outcome has remained at a dismal 5% over the last 2 decades, and while the treatment of advanced OSCC requires an integrative multimodal approach, including surgical resection, radiation therapy, and cisplatin-based chemotherapy, surgery remains the treatment modality of choice, howbeit beleaguered with several foci of controversy pertaining to preoperative work-up, management of the primary tumors, and initiation of (neo)adjuvant therapy [1,2,3,4,5,6,7]. More so, since the response of tumors to neoadjuvant chemotherapy is predictive of probable complete response to radiation therapy or absence of disease recurrence, strong evidence of chemosensitivity or inherent ability to impair innate or adaptive therapy-resistance is pivotal in deciding type of conservative treatment, thus, necessitating the identification of novel molecular targets or predictors of tumor response to treatment, and/or development of new therapeutic strategies to overcome primary resistance.

Cell-cell adhesion and communication molecules, such as the cadherins are constitutively essential for facilitation of cellular signaling pathways and maintenance of physiological functions. Impaired regulation and resultant aberration in the expression and/or activity of cadherins have been associated, or better put, implicated in several pathological condition, including nephropathies, autoimmune diseases, and malignancies [27,28,29]. “As a member of the Ca^2+^-dependent adhesion super-family, FAT proteins were first described in the 1920s as an inheritable lethal mutant phenotype in *Drosophila*, consisting of four member proteins, FAT1, FAT2, FAT3, and FAT4, all of which are highly conserved in structure [29].” Against the background of conflicting roles ascribed to FAT1 in different malignancies, this present study demonstrates that: (i) high FAT1 expression drives HNSC, defines high risk population and is associated with poor survival, as indicated by strong positive correlation with disease progression and poor clinical outcome in patients with HNSC. Moreover, the first time to the best of our knowledge we showed that (ii) FAT1 is aberrantly-expressed in OSCC clinical samples and cell lines, exhibiting strong association with an activated LRP5 signalosome, dampened oxidative stress, and poor prognosis in patients with OSCC, while the (iii) loss-of-FAT1 function impairs OSCC cell proliferation, enhances cell death, and attenuates the oncogenicity and metastatic phenotypes of OSCC cells. In addition, we provided pre-clinical evidence that (iv) the molecular targeting of FAT1 re-sensitizes cisplatin-resistant OSCC cells to cisplatin through deregulation of LRP5/WNT2/GSS signaling axis and enhanced oxidative stress.

Findings documented herein complement and are consistent with accruing evidence of the tumor initiating and tumor driving roles of FAT1 in various cancer types, such as the demonstrated role of FAT1 as an upstream master regulator of HIF1α expression and activity, and its ability to enhance the invasiveness of GBM cells under hypoxic conditions [19], while conversely, the small interfering RNA-mediated suppression of FAT1 induced upregulated expression of the tumor-suppressor programmed cell death 4 (PDCD4) gene, with concomitant inhibition of c-Jun phosphorylation and activator protein-1 (AP-1) transcription, resulting in reduced migration and invasiveness of GBM cells [30]. Similarly, there is evidence that relative to its marginal expression in normal human colorectal cancer (CRC) samples, FAT1 is broadly expressed in primary and metastatic CRC cells, mainly accumulating in the plasma membrane, regardless of *KRAS* and *BRAF* mutations, and positively correlated with enhanced cell invasiveness, conversely, when targeted by the FAT1-specific monoclonal antibody, mAb198.3, reduction in cancer growth ensues in colon cancer xenograft model, in vivo [31]. Moreover, aside its ability to induce metastasis and progression of hepatocellular carcinoma upon interaction with POU2F1/OCT1 [32], FAT1 expression has been shown to be up-regulated in metastatic gastric cancer, and patients with FAT1^high^ gastric cancer had worse prognosis [33].

Regardless of the aforementioned consistencies between our present findings and several other published works, we do acknowledge our findings contradict those of Lin et al. [17], suggesting FAT1 acts as a tumor suppressor, with lower FAT1 protein expression bearing significant correlation with lymph node metastasis, lymphovascular permeation, tumor recurrence, and shorter disease-free survival (DFS) in patients with HNSC. We cannot fully explain this contradiction, however, we cautiously attribute this to tumor heterogeneity and/or cohort constitution, based on demonstrated association of high expression of FAT1 with high grade cancerous cells and low FAT1 with low grade cancerous cells in previous works [19,30], as observed in the present work with the use of the HSC-3 and SAS cells which are poorly differentiated human squamous carcinoma of the tongue cells with high lymph node metastasis potential [34,35,36], as well as with our use of OSCC cohorts with more high grade, advanced stage or metastatic cases. As rightly noted by Soussi and Wiman for p53 (TP53) [37], it is not impossible that while the standard criteria for definition of various cancer genes may confine the tumor protein FAT1 to the role of a tumor suppressor, accruing evidence across multiple cancer types with diverse histology, indicate that FAT1 does indeed act as an oncogene.

The present study thus provides rationale to look outside the box of classical and traditional classification of FAT1 as a tumor suppressor, by highlighting various oncogenic properties that make FAT1 a putative therapeutic target that should not be underestimated in OSCC. Consistent with rationalizations on genes with both oncogenic and tumor-suppressor functions by Shen et al. [38], it is also probable that the function-altering mutations in FAT1 are the main driving force in the FAT1-facilitated oncogenesis and cisplatin resistance in OSCC documented herein. Interestingly, like TP53, the mode of FAT1 inactivation is quite unique, compared with most tumor suppressors; ~22% and 75% of FAT1 genomic alterations are missense and truncating/nonsense mutations, respectively, both of which facilitate the synthesis of a stable mutant FAT1 protein which accumulates in the plasma membrane and nucleus of the aggressive and/or cisplatin-resistant OSCC cells. Comparatively, this high frequency of amino acid substitution (missense) or premature termination of translation (truncating/nonsense) is highly analogous with various cancer types, regardless of the difference in mutation spectrum [37]. Finally, consistent with Muller’s exposition on the nature and causes of gene mutations, based on the classification of mutations hinged on genophenotypic analyses [39], genomic alterations in FAT1 are akin to the ‘amorph’ or ‘hypomorphic’ mutation which are more characteristic of so-called ‘tumor suppressors’, wherein the tumor suppressor function is totally impaired or a partial reduced, resulting in its acquisition of oncogenicity and ability to drive cancer. While it is conceivable that missense and nonsense/truncating mutations resulting in true amorphic variants elicit complete loss of tumor suppressor function, in many instances it is difficult to exclude residual activities that result in heterogeneous hypomorphic variants with context-dependent functional duality as documented in the present study for the protocadherin FAT1. This rationalization based on tumor heterogeneity and mutational status are therapeutically relevant as they go beyond the initial biological function ascribed to FAT1, and can inform discovery or development of novel anti-OSCC therapeutic strategies with high efficacy.

Contextually, our FAT1 findings are, at least in part, corroborated by data indicating that the aberrant expression of the male-specific protocadherin-PC (PCDH-PC) in prostate cancer cells facilitate their acquisition of an apoptosis-evading and hormone therapy-resistant phenotype through enhanced nuclear accumulation of β-catenin and increased WNT-signaling [40]. Also, the observed strong co-expression of FAT1, molecular components of the LRP5 signalosome, namely LRP8, GSK3β, WNT2, β-catenin, casein kinase 1 gamma 1 (CSNK1G1), CSNK1G2, CSNK1G3, AXIN1, caveolin (CAV)1, CAV2, and glutathione (GSS), with weak expression of glutathione-disulfide reductase (GSR) in patients with OSCC, compared with the dysplasia or normal control group, is partially in concordant with reports demonstrating a positive correlation between expression of CAV1 and LRP5-analogous LRP6 in human primary and metastatic prostate cancer tissues, and that the interaction between CAV1 and LRP6 plays an important role in the regulation of Wnt/β-catenin signaling [41].

Moreover, our finding demonstrating that the ablative targeting of FAT1 re-sensitizes cisplatin-resistant OSCC cells to cisplatin through suppressed glutathione (GSH) expression, enhanced oxidative stress and deregulation of LRP/Wnt signaling, become therapeutically relevant when put in the context of contemporary knowledge that while resistance to chemotherapeutics constitute a major impediment to treatment success, cisplatin-resistance can be overcome by inhibition of glutathione S-transferase (GSTs), in vitro and in vivo, where the activity of GST is dependent on the steady production or availability of glutathione (GSH) [25,42,43], and deregulated GSH homeostasis is implicated in the pathogenesis and progression of several human diseases including malignancies, especially as impaired GSH production, or decreased GSH/glutathione disulphide (GSSG) ratio, results in enhanced susceptibility to oxidative stress, which in turn is culpable in cancer progression, where elevated GSH levels augment the antioxidant capacity and resistance to oxidative stress characteristic of many cancerous cells, including OSCC [25]. Thus, consistent with Matés et al’s exposition on the implication of oxidative stress for cell proliferation, apoptosis and carcinogenesis [44], the present study expounds the role of FAT1 in the modulation of oral carcinogenesis, oxidative stress and cisplatin resistance via deregulation of the LRP5/WNT2/GSS signaling axis, wherein all molecular factors alluded to in the study are molecular effectors of the crosstalk between WNT/β-catenin and GSH oxidative stress signaling pathways.

Put together, as summarized in the Graphical abstract**,** this present study uncovers a new role for FAT1 in OSCC oncogenesis and cisplatin resistance, with some mechanistic insight into this oncogenic role, thereby highlighting the functional duality of FAT1 in OSCC. Our findings provide pre-clinical evidence for the therapeutic exploitation of FAT1 ambivalence especially in the treatment of metastatic and/or cisplatin-resistant disease.

## 4. Materials and Methods

### 4.1. Reagents and Drugs

Cisplatin (*cis*-diamineplatinum (II) dichloride, #479306, ≥99.9% trace metal basis) was purchased from Sigma-Aldrich, Inc (St. Louis, MO, USA). Stock solutions of 100 mM in sterile ddH_2_O were stored in the dark at 4°C, respectively, until use. Antibodies against FAT1 (FAT-1 3D7/1, #sc-53283), PCNA (F-2, #sc-25280), and GSH (D8, #sc-52399) were obtained from Santa Cruz Biotechnology Inc. (Santa Cruz, CA, USA), while Bax (D2E11, #5023), Bcl2 (124, #15071), LRP5 (D80F2, #5731), GSK3β (D5C5Z, #12456), p-GSK3β (5B3, #9323), active β-catenin (D13A1, #8814), and β-actin (D6A8, #8457) were purchased from Cell Signaling Technology (CST, Beverly, MA, USA).

### 4.2. OSCC Tissue Specimens

We obtained 21 matched OSCC and adjacent non-tumor tissue samples as kind gift from Dr. Chun-Shu Lin, from the National Defense Medical Centre, Tri-Service General Hospital OSCC tissue bank, following ethical approval for their use from the Institutional Review Board of the Tri-Service General Hospital (TSGHIRB 2-102-05-125). Requirement for patients’ signed informed consent was waived because tissue samples were obtained retrospectively from the Tri-Service General Hospital OSCC archive.

### 4.3. Cell Lines and Cell Culture

Adult Human Primary Normal Gingival Fibroblast cells (HGF, ATCC® PCS-201-018™) obtained from the American Type Culture Collection (Manassas, VA, USA), was cultured in Fibroblast growth medium (#116-500, Sigma-Aldrich Inc.). Human OSCC cell line with high metastatic potential HSC-3 (#JCRB0623) and SAS (#JCRB0260) were purchased from the National Institutes of Biomedical Innovation, Health and Nutrition (NIBIOHN)-Japanese Collection of Research Bioresources (JCRB) Cell Bank (Osaka, Japan), and cultured in Gibco™ Dulbecco’s Modified Eagle Medium (DMEM, #11995065, Thermo Fisher Scientific Inc., Bartlesville, OK, USA), supplemented with 10% fetal bovine serum (FBS, #26140079) and 1% penicillin-streptomycin at 37 °C, in 5% humidified CO_2_ incubator. Cells used in the present study were all ≤ passage number 3 (P.3) and were sub-cultured when 90% confluent and media changed every 48 h.

### 4.4. Mutation Status Analyses

For bioinformatics analyses of the mutational status of patients with OSCC, we accessed and re-analyzed publicly available databases, namely, the TCGA HNSC cohort using the Catalog of Somatic Mutations In Cancer (COSMIC) (https://cancer.sanger.ac.uk/cosmic/study/overview?study_id=627), DriverDBv3 database for human cancer driver gene research (http://driverdb.tms.cmu.edu.tw/), and cBioPortal for cancer genomics (https://www.cbioportal.org/study/summary?id=hnsc_tcga).

### 4.5. Immunohistochemistry

After deparaffinizing the paraffin-embedded OSCC tissue sections in xylene, antigen retrieval was performed by rehydration in decreasing concentration of ethanol and heating in citrate buffer (pH 6.0). After washing, the tissue slides were blocked with 3.0% hydrogen peroxide (H_2_O_2_) and 10% goat serum and then incubated with primary antibody against FAT1 (1:400, FAT-1 3D7/1, #sc-53283; 

Santa Cruz Biotechnology Inc.) at 4 °C overnight. Thereafter, the slides were sequentially incubated with appropriate biotinylated secondary antibody, streptavidin-horseradish peroxidase (HRP) complex and diaminobenzidine (DAB). The stained slides were then counterstained with hematoxylin, dehydrated, and mounted, followed by visualization and imaging under light microscope.

### 4.6. shFAT1 Transfection and Establishment of Stable Knockdown Cell Lines

The small hairpin RNA (shRNA) specifically targeting human FAT1 (FAT1 shRNA Plasmid (h); #sc-88872-SH) obtained from Santa Cruz Biotechnology Inc. was transfected into HSC3 or SAS cells grown in 6-well plates to 60% confluence using Lipofectamine 2000 reagent (Invitrogen) according to the manufacturer’s instructions. For puromycin selection of stably transfected clones, 48 h after shFAT1 transfection, medium was aspirated and replaced with fresh medium containing 5 μg/mL puromycin and incubated for another 48 h in humidified 5% CO_2_ atmosphere incubator at 37 °C. Thereafter, shFAT1-transfected cells were harvested for qRT-PCR or western blot analysis.

### 4.7. Cell Migration and Invasion Assays

Cell migration capability was evaluated using the wound healing assay. Briefly, wild-type or shFAT1-transfected HSC-3 or SAS cells were seeded into 6-well plates (Corning Inc., Corning, NY, USA) containing complete growth media supplemented with 10% FBS, and cultured to ≥98% monolayer confluency. The cell monolayers were scratched with sterile yellow pipette tips to denude the culture wells. Images of cell migration were captured at the 0 and 24 h time-points after denudation, under a microscope with a 10× objective lens, and later analyzed with the NIH ImageJ software (https://imagej.nih.gov/ij/download.html).

For invasion assay, using 24-well plate matrigel Transwell® systems, we seeded 3 × 10^4^ cells into the upper chamber of the insert (BD Bioscience, pore size = 8 μm) containing FBS-free media, while the lower chamber contained 10% FBS-supplemented media. After incubation for 24 h, all media were carefully discarded, non-invaded cells in the upper surface of the insert were removed carefully with sterile cotton swipes, while invaded cells on the underside of the membrane were stained with crystal violet dye after fixture with 3.7% formaldehyde, and then the average number of invaded cells were determined under microscope, from at least five non-overlapping visual fields selected randomly.

### 4.8. Western Blot Analysis

Cultured HSC-3 or SAS cells were collected, lysed, and the cell protein lysates heated for 5 min, before immunoblotting. Thereafter, blots were blocked with 5% skimmed milk in TBST for 1 h to avoid non-specific binding, incubated with monoclonal primary antibodies against FAT1 (1:1000), PCNA (1:1000), Bax (1:1000), Bcl2 (1:1000), LRP5 (1:1000), GSK3β (1:1000), p-GSK3β (1:1000), active β-catenin (1:1000), and β-actin (1:2000) at 4 °C overnight, and then the polyvinylidene difluoride (PVDF) membranes were washed thrice with TBST, incubated with horseradish peroxidise (HRP)-labeled secondary antibodies at room temperature for 1h, and then washed thrice with TBST again before band visualization with the Pierce™ enhanced chemiluminescence (ECL) Western blotting substrate (#32106, Thermo Fisher Scientific Inc., Bartlesville, OK, USA) and imaging using the BioSpectrum Imaging System (UVP, Upland, CA, USA).

### 4.9. Cell Viability and Proliferation Assay

Wild-type or shFAT1-transfected HSC-3, SAS and HGF cells (2 × 10^4^ cells/well) were seeded in triplicates in 24-well plates containing 500 μL supplemented media, and incubated in humidified 5% CO_2_ atmosphere incubator at 37 °C for 48 h. Cell proliferation were assessed by sulforhodamine B (SRB) colorimetric assay as previously described [45]. Assay was performed four times in triplicates, with the wild-type OSCC cells served as control. Cell proliferation/viability was determined by absorbance based on optical density (OD) measured at 495 nm wavelength, using the SpectraMax microplate reader (Molecular Devices, Kim Forest Enterprises Co., Ltd, New Taipei, Taiwan).

### 4.10. Establishment of Cisplatin-Resistant (CispR) OSCC Cells

The human OSCC cell lines HSC-3 and SAS were cultured in DMEM supplemented with 10% fetal bovine serum (FBS), and 1% penicillin/ streptomycin (P/S) and were incubated in a humidified 37 °C incubator with 5% CO_2_. To establish cisplatin-resistant (CispR) human OSCC cells, parental human OSCC cell lines HSC-3 or SAS were gradually exposed to increasing concentrations of cisplatin for 6 months until only 50% of the original seeded cells which are tolerant to cisplatin (IC_50_) were left. After determination of and reaching IC_50_, the culture media was changed and the cisplatin-tolerant/resistant HSC-3 or SAS cells were then cultured in media containing cisplatin at IC_20_ concentration for subsequent cultures (SAS CispR: 3.8 μM; HSC-3 CispR: 3.1 μM). Before using in subsequent assays, the CispR cells were cultured in fresh cisplatin-free complete growth media for 72 h. The cisplatin-resistant phenotype of the CispR OSCC cells was routinely checked by SRB assay every 2 months.

### 4.11. Measurement of Intracellular Glutathione (GSH) Levels

1.5 × 10^6^ wild-type or cisplatin-resistant HSC-3 cells were seeded into Corning®10-cm cell culture dishes (Corning Inc.) and incubated in humidified 5% CO_2_ atmosphere incubator at 37 °C for 48 h. After deproteinization with 5% 5-sulfosalicylic acid dihydrate ACS reagent (#247006, ≥99% HPLC, Sigma-Aldrich Inc.). Intracellular GSH levels was determined using the Glutathione Assay Kit (#CS0260, Sigma-Aldrich Inc.), following the manufacturer’s instructions.

### 4.12. Real-time Polymerase Chain Reaction (qRT-PCR)

Following the isolation of total RNA from parental or cisplatin-resistant (CispR) HSC-3 or SAS cells, using the RNeasy Mini Kit (#74106, Qiagen Inc., Germantown, MD, USA), polymerase chain reaction (PCR) mixtures containing the primers and fluorogenic probe mix were prepared using the GeneRead qPCR SYBR® Green Master Mix (#180820, Qiagen Inc.). Amplification reactions were performed in triplicate using 20 ng cDNA in the Bio-Rad C1000 real-time PCR system (Bio-Rad, Cambridge, MA, USA) using the condition: 95 °C for 3 min, 35 cycles at 95 °C for 15, 60 °C for 30 s, 72 °C for 30 s, and 72 °C for 10 min. The specific primers used were as follows: FAT1 (forward) 5’-ACTTCTAGAGCCACCATGGTCGCTCATTCC-3’, (reverse) 5’-ACTGCTAGCTTACTTGGCCTTT GCCTTCT-3’, and β-actin (forward) 5’-GACCTCTATGCCAACACAGT-3’, (reverse) 5’-AGTACTT GCGCTCAGGAGGA-3’. Results were analyzed and all values were normalized to the levels of GAPDH.

### 4.13. Statistical Analysis

All experiments were performed at least three times in triplicates, and data presented represent mean ± standard deviation (SD). Comparison between two groups was done using 2-sided Student’s t-test, and the 1-way analysis of variance (ANOVA) used for comparison between ≥3 groups. All statistical analyses were performed utilizing the GraphPad Prism version 7.00 for Windows (GraphPad Software Inc., La Jolla, CA, USA). *p*-value < 0.05 was considered statistically significant.

## 5. Conclusions

In conclusion, as summarized in the Graphical abstract, this present study uncovers a new role for FAT1 in OSCC oncogenesis and cisplatin resistance, with some mechanistic insight into this oncogenic role, thereby highlighting the functional duality of FAT1 in OSCC. Our findings provide pre-clinical evidence for the therapeutic exploitation of FAT1 ambivalence especially in the treatment of metastatic and/or cisplatin-resistant disease.

## Figures and Tables

**Figure 1 cancers-11-01883-f001:**
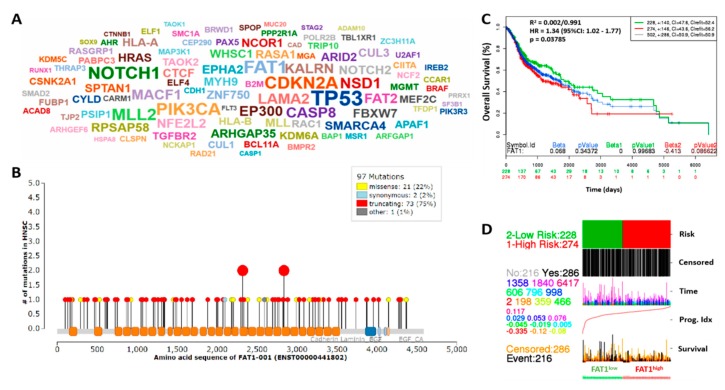
High FAT1 expression drives HNSC, defines high risk population and is associated with poor survival. (**A**) Driver cloud showing the most recurrently mutated cancer driver genes in the TCGA-HNSC dataset; The gene symbol size corresponds to count of samples with protein-affecting mutations (PAMs). (**B**) Mutations distribution along FAT1 protein sequence. (**C**) KM plot of the FAT1 expression-stratified overall survival of the TCGA HNSC June 2016 cohort (*n* = 502). (**D**) Correlative functional analysis of FAT1 expression, risk, prognosis and survival in the TCGA-HNSC cohort.

**Figure 2 cancers-11-01883-f002:**
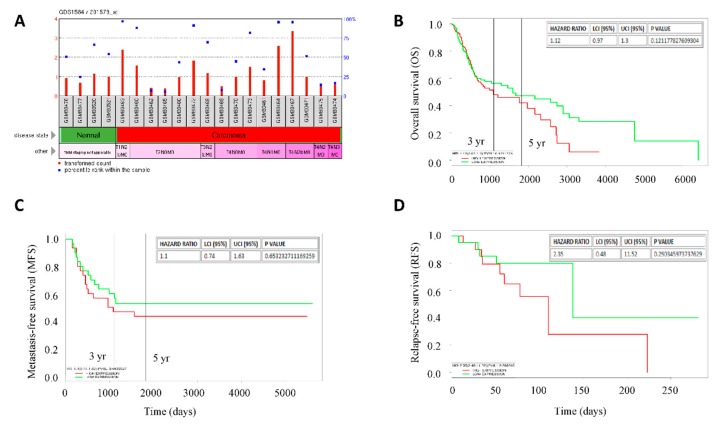
High FAT1 expression is positively correlated with disease progression and poor clinical outcome in patients with HNSC. (**A**) Histograms of the stage-associated differential expression of FAT1 in the GPL96/GDS1584 gene expression profiling by array data-set, transformed count, 20 samples. Kaplan-Meier plots of the correlation between differential FAT1 expression and (**B**) overall survival of OSCC cohort of the TCGA-HNSC, *n* = 291, (**C**) metastasis-free survival of patients with HNSC, *n* = 89 using the E-MTAB-1328 dataset, and (**D**) relapse-free survival of patients undergoing surgery or biopsy for HNSC at University of North Carolina, Chapel Hill and Vanderbilt University using the GSE10300 dataset, *n* = 44. High/Low expression bifurcation was based on median FAT1 expression in all cases.

**Figure 3 cancers-11-01883-f003:**
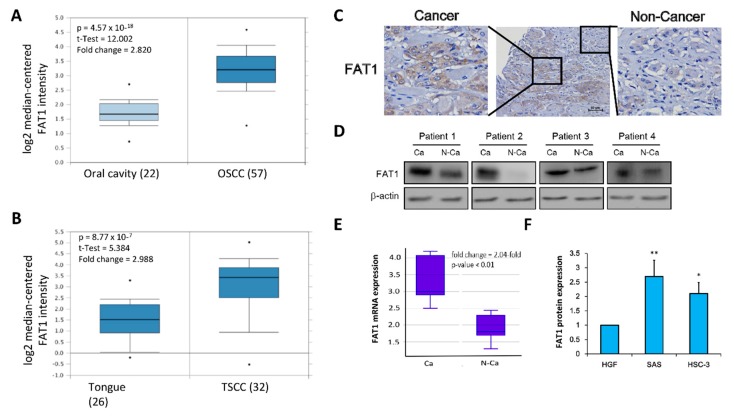
FAT1 is aberrantly-expressed in OSCC clinical samples and cell lines. Box and whiskers plots of the differential expression of FAT1 in (**A**) Peng Head-Neck OSCC cohort (*n* = 79), and (**B**) Estilo Head-Neck TSCC cohort (n = 58) (**C**) Representative immunohistochemical staining of FAT1 in a human OSCC tissue section showing both cancerous and neighbouring non-cancerous areas (Scale bars, 50 μm). (**D**) Representative western blot images of the differential expression of FAT1 in cancerous and non-cancerous tissues from our in-house OSCC cohort. (**E**) Box plot showing differential expression of FAT1 mRNA level in paired human OSCC tissue (*n* = 12) and non-cancerous tissue (*n* = 12). (**F**) Histogram of the basal FAT1 mRNA expression in primary human gingival fibroblast PCS-201-018 cells, poorly differentiated SAS, metastatic HSC-3 and OSCC cell lines. β-actin served as loading control. * *p* < 0.05, ** *p* < 0.01.

**Figure 4 cancers-11-01883-f004:**
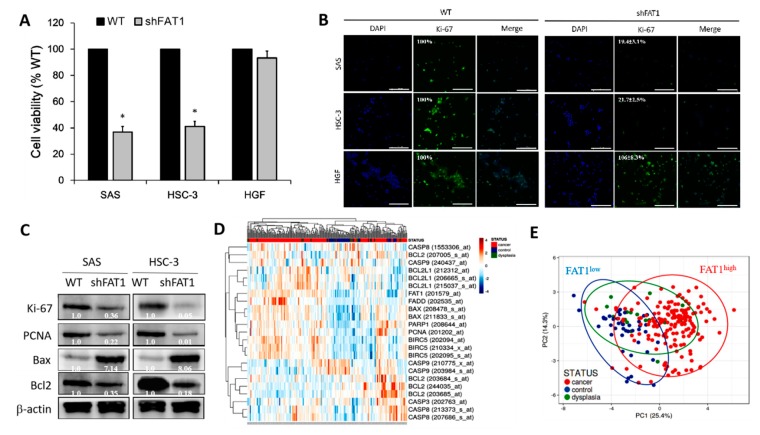
Loss-of-FAT1 function impairs OSCC cell proliferation and enhances cell death. (**A**) Images (upper panel) and graphical representation (lower panel) of the effect of shFAT1 on the viability/proliferation of SAS, HSC-3 or HGF cells. (**B**) Representative Ki-67 protein immunofluorescence staining images in wild-type and shFAT1 SAS, HSC-3 or HGF cells. DAPI served as nuclear marker. (**C**) Representative western blot images of the differential expression of Ki-67, PCNA, Bcl2, and Bax proteins in wild-type or shFAT1 SAS and HSC-3 cells. (**D**) Heatmap of FAT1, PCNA, BIRC5, BCL2, BCL2L1, FADD, BAX, PARP1, CASP3, CASP8, and CASP9 gene expression profile of OSCC cohort (*n* = 229 samples, 54675 genes) using the A-AFFY-44, AFFY_HG_U133_PLUS_2, E-GEOD-30784 dataset. Rows are centered; unit variance scaling is applied to rows. Both rows and columns are clustered using correlation distance and average linkage. 23 rows, 229 columns. (**E**) Principal component analysis of same E-GEOD-30784 OSCC cohort. Unit variance scaling is applied to rows; SVD with imputation is used to calculate principal components. X and Y axis show principal component 1 and principal component 2 that explain 25.4% and 14.3% of the total variance, respectively. Prediction ellipses are such that with probability 0.95, a new observation from the same group will fall inside the ellipse. *n* = 229 data points. β-actin served as loading control. * *p* < 0.05.

**Figure 5 cancers-11-01883-f005:**
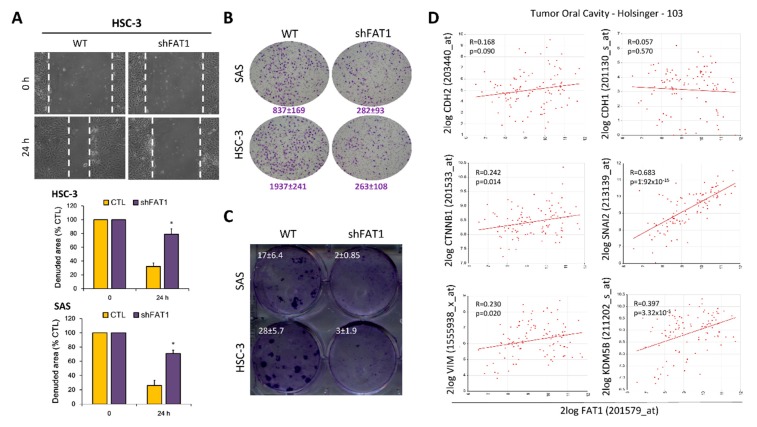
Loss-of-FAT1 function attenuates the oncogenicity and metastatic phenotype of OSCC cells. (**A**) wound healing migration assay (**B**) Matrigel invasion assay (**C**) Colony formation assay (**D**) Graphical representations of correlation between FAT1, CDH1, CDH2, CTNNB1, SNAI2, VIM, and KDM5B using the Holsinger GPL570/GSE42743/GSM1049079 Oral Cavity Cancer cohort (*n* = 103). * *p* < 0.05.

**Figure 6 cancers-11-01883-f006:**
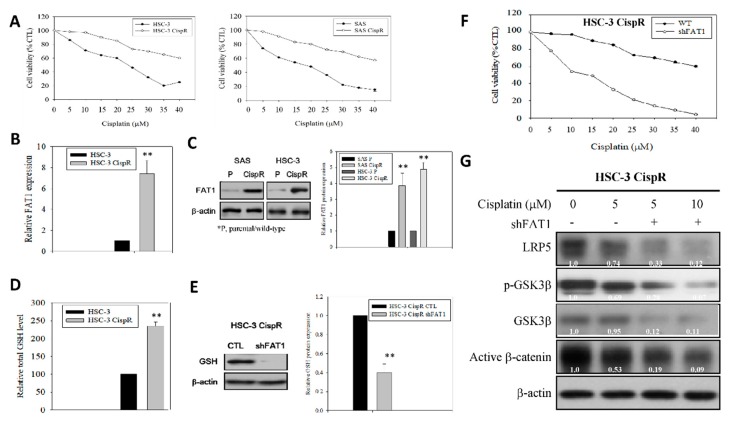
Ablative targeting of FAT1 re-sensitizes cisplatin-resistant OSCC cells to cisplatin through deregulation of LRP/Wnt signaling. (**A**) Graphical representation of the effect of 48h cisplatin treatment on wild-type or cisplatin-resistant HSC-3 and SAS cells. (**B**) Histogram of the differential expression of FAT1 mRNA in wild-type or cisplatin-resistant HSC-3 from qRT-PCR. (**C**) Western blot images (upper) and graphs (lower) of the differential expression of FAT1 protein in wild-type or cisplatin-resistant HSC-3 and SAS cells. (**D**) The wild-type or cisplatin-resistant HSC-3 were seeded at a density of 1 x 106 cells per 10-cm dish. After culture overnight, the cells were collected and the total cellular GSH levels were analyzed by GSH assay kit. (**E**) Representative western blot images (upper) and graphs (lower) showing the expression of GSH in parental control or shFAT1 HSC-3 CispR cells. (**F**) Graph showing effect of shFAT1 on the sensitivity of parental control or shFAT1 HSC-3 CispR cells to 48 h cisplatin treatment. (**G**) Representative western blot images of the effect of cisplatin alone or with shFAT1 on the expression of LRP5, p-GSK3β, GSK3β, and active β-catenin in HSC-3 CispR cells. β-actin is loading control. * *p* < 0.05, ** *p* < 0.01; CispR, cisplatin resistant; *p*, parental; GSH, glutathione.

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
