# Peer review of "Targeting FAT1 Inhibits Carcinogenesis, Induces Oxidative Stress and Enhances Cisplatin Sensitivity through Deregulation of LRP5/WNT2/GSS Signaling Axis in Oral Squamous Cell Carcinoma"

_cancers, 2019, doi:10.3390/cancers11121883_

Round 1

Reviewer 1 Report

please see the attached file.

Author Response

Response to Reviewers:

Point-by-point responses to reviewer’s comments:

We would like to thank the reviewer for the thorough reading of our manuscript as well as their valuable comments. We believe all comments are borne out of good faith, and thus, have tried to address their comments conscientiously and feel that they have further improved the readability and appeal of our work, as well as strengthened the manuscript. Below are our point-by-point responses.

Q1.1.    In this study, the author tried to investigate the biological role of FAT1 in HNSC. The data showed that FAT1 depletion caused the decrease of cell proliferation and enhanced cell apoptosis. FAT1 associated with CDH1, CDH2, CTNNB1, SNAI2, VIM, and KDM5B to modulate oncogenicity and metastatic phenotype of OSCC cells. In addition, FAT1 expression re-sensitized to cisplatin in OSCC cells through oxidative stress and deregulation of LRP/Wnt signaling. This study is crucial and provided an information that the significance of FAT1 in OSCC progression. To improve the quality of manuscript, some issues were suggested:

A1.1.    We thank the reviewer for taking time to read our manuscript and for the critiques and suggestions made in order to help us improve the quality of our work. In this revised manuscript, we have made effort to make address all the comments and suggestions.

Q1.2.     In figure 1A, 131 mutated drivers were determined from TCGA HNSC database by using bioinformatics. The analyzed approaches should be presented in the materials and methods. In addition, in figure 1C, what did the "blue line" and x-axis mean?

 A1.2.   We thank the reviewer for this suggestion. We have now included a section on the analyses of mutations in cancer as requested by the reviewer in our revised manuscript. Please kindly see our revised Materials and Methods section, Page 4, Lines 134-140:

2.4. Mutation status analyses

For bioinformatics analyses of the mutational status of patients with OSCC, we accessed and re-analyzed publicly available databases, namely, the TCGA HNSC cohort using the Catalog of Somatic Mutations In Cancer (COSMIC) (https://cancer.sanger.ac.uk/cosmic/ study/overview?study_id=627), DriverDBv3 database for human cancer driver gene research (http://driverdb.tms.cmu.edu.tw/ ), and cBioPortal for cancer genomics (https://www.cbioportal.org/ study/summary?id=hnsc_tcga).

Regarding figure 1C, the "blue line" shows the median overall survival rate, regardless of variability in FAT1 expression; and for x-axis, we have now included the legend. Please kindly see our updated Figure 1C and its legend, Pages 6-7, Lines 238-247:

Figure 1. High FAT1 expression drives HNSC, defines high risk population and is associated with poor survival. (A) Driver cloud showing the most recurrently mutated cancer driver genes in the TCGA-HNSC dataset; The gene symbol size corresponds to count of samples with protein-affecting mutations (PAMs). (B) Mutations distribution along FAT1 protein sequence. (C) KM plot of the FAT1 expression-stratified overall survival of the TCGA HNSC June 2016 cohort (n =502). (D) Correlative functional analysis of FAT1 expression, risk, prognosis and survival in the TCGA-HNSC cohort.

Q1.3.    In figure 2A, author demonstrated that FAT1 expression was associated with nodal involvement in OSCC database. According to the result of figure 2A, this statement was not suitable. Please provide statistical evidence to further support the findings.

A1.3.    We thank the reviewer for this comment. As requested, we have now provided ‘statistical evidence to further support the findings’ in our revised manuscript. Please kindly see our updated Supplementary Figure S4 and its legend, Pages 20-21, Lines 717-727:

Supplementary Figure S4. High FAT1 expression is positively correlated with nodal        involvement and poor clinical outcome in patients with HNSC.  Boxplot and whiskers plots of the correlation between (A) clinical node (N) stage, or (B) number of lymph-nodes positive    by IHC, and the differential expression of FAT1 in TCGA-HNSC cohort. N, nodal; IHC,                               immunohistochemistry. Kaplan-Meier plots of the correlation between differential FAT1    expression and (C) relapse-free survival in the Chung Parker Head and Neck GSE10300  cohort, n = 44, or  (D) overall survival in the TCGA-HNSC June 2016 cohort, n = 502.

Q1.4.    In figure 2B to D, the data indicated that high FAT1 expression was associated with worse overall survival, shorter metastasis-free survival, and shorter relapse-free survival than that in HNSC patients with low FAT1 expression. According to the statistic results presented by authors, the p value was not significant. It was confused that if FAT1 expression was associated with poor clinical outcome in patients with HNSC. Please provide statistical evidence to further support the findings. Authors should describe the results according to their findings.

A1.4.    We sincerely thank the reviewer for this comment, however, we politely point out that while a significant p-value may suggest an experimental arm out-performs a control arm, it does not and indeed cannot indicate the magnitude of the difference. It is this inherent lack of ability to reveal or inform magnitude of difference or change that renders p-value irrelevant in many assays, including the one alluded to by the erudite reviewer. It is important to remember that “statistical significance does not equal or translate to biological significance”; A very high level of statistical significance (p >> 0.05) may mean nothing in practical sense. As rightly ut by Prof. Warren S. Browner, “A p-value simply reflects the probability of a set of findings (or ones more extreme) under the null hypothesis that there was no difference in the groups being compared. P-values (and confidence intervals, for that matter) do not account for the likelihood of the hypothesis being tested and cannot distinguish “true” from “false” results. They are purely data-based.” May we humbly refer the reviewer to the follow:

Halsey et al. (2015) The fickle P value generates irreproducible results. Nature Method, 12(3), 179–185. Sullivan, G. M., & Feinn, R. (2012) Using Effect Size or Why the P Value Is Not Enough. Journal of Graduate Medical Education, 4(3), 279–82. Browner, W.S. (2003) The Reliability of P Values. Science, 301(5630), 167-168.

             However to allay the reviewer’s concern, we have now included statistical evidence to further support our findings that FAT1 expression is associated with poor clinical outcome in patients with HNSC. Please kindly see our updated Supplementary Figure S4 and its legend, Pages 20-21, Lines 717-727:

Supplementary Figure S4. High FAT1 expression is positively correlated with nodal        involvement and poor clinical outcome in patients with HNSC.  Boxplot and whiskers plots of the correlation between (A) clinical node (N) stage, or (B) number of lymph-nodes positive    by IHC, and the differential expression of FAT1 in TCGA-HNSC cohort. N, nodal; IHC,                               immunohistochemistry. Kaplan-Meier plots of the correlation between differential FAT1    expression and (C) relapse-free survival in the Chung Parker Head and Neck GSE10300  cohort, n = 44, or  (D) overall survival in the TCGA-HNSC June 2016 cohort, n = 502.

Q1.5.    In figure 3F, the protein expression profiles of FAT1 among these cell lines should be presented by Western blotting. Moreover, the figure legend (3F) was not matched the figure of 3F.

A1.5.    We thank the reviewer for this comment. We have now corrected the figure legend to match the data provided. Please kindly see our revised Figure 3F legend, Page 9, Lines 301-309:

Figure 3. FAT1 is aberrantly-expressed in OSCC clinical samples and cell lines. (A) Peng Head-Neck OSCC cohort (n = 79)  (B) Estilo Head-Neck TSCC cohort (n = 58) (C) Representative immunohistochemical staining of FAT1 in a human OSCC tissue section showing both cancerous and neighbouring non-cancerous areas (Scale bars, 50 μm). (D) Representative western blot images of the differential expression of FAT1 in cancerous and non-cancerous tissues from our in-house OSCC cohort. (E) Box plot showing differential expression of FAT1 mRNA level in paired human OSCC tissue (n=12) and non-cancerous tissue (n=12). (F) Histogram of the basal FAT1 mRNA expression in primary human gingival fibroblast PCS-201-018 cells, poorly differentiated SAS, metastatic HSC-3 and OSCC cell lines. b-actin served as loading control. *p<0.05, **p<0.01, ***p<0.001.

Q1.6.    In figure 4A, what is the "wild type" mean? What was the sequence of shFAT1? Please show it in the materials and methods section. FAT1 was also observed in HGF (normal gingival fibroblast) cells; however the proliferation/viability of HGF was barely influenced while FAT1 gene was suppressed. Why? Please explain it.

A1.6.    We appreciate the reviewer comments. As is the convention in many published works in peer-reviewed journals, by wild-type (WT), we refer to the parental cell line not transfected with shRNA targeting FAT1.

Regarding the sequence of shFAT1, we humbly remind the reviewer that this a commercial small hairpin RNA (shRNA) specifically targeting human FAT1 (FAT1 shRNA Plasmid (h)) obtained from Santa Cruz Biotechnology Inc. (Santa Cruz, CA, USA), with catalog number #sc-88872-SH. Therefore, we can not provide the requested sequence. Please kindly see our revised Materials and Methods section, Page 4, Lines 151-159:

2.6. shFAT1 transfection and establishment of stable knockdown cell lines

The small hairpin RNA (shRNA) specifically targeting human FAT1 (FAT1 shRNA Plasmid (h); #sc-88872-SH) obtained from Santa Cruz Biotechnology Inc. (Santa Cruz, CA, USA ) was transfected into HSC3 or SAS cells grown in 6-well plates to 60% confluence using Lipofectamine 2000 reagent (Invitrogen) according to the manufacturer’s instructions. For puromycin selection of stably transfected clones, 48 h after shFAT1 transfection, medium was aspirated and replaced with fresh medium containing 5 mg/mL puromycin and incubated for another 48 h in humidified 5% CO2 atmosphere incubator at 37 °C. Thereafter, shFAT1-transfected cells were harvested for qRT-PCR or western blot analysis.

As rightly noted by the reviewer, “the proliferation/viability of HGF was barely influenced while FAT1 gene was suppressed”. This is consistent with the expected ‘ability to spare normal or non-malignant cells’ and ‘selectivity for cancerous cells’ of any effective anticancer strategy. Please kindly see our revised Results section, Page 9, Lines 311-341:

3.4. Loss-of-FAT1 function impairs OSCC cell proliferation and enhances cell death.

As with all our works, since the objective is to proffer solution and not as much focus on the problem, we investigated if and to what degree the loss-of-FAT1 function negatively affects the survival and proliferation of OSCC cells.  Our results indicate that shFAT1 transfection markedly suppressed the proliferation cum viability of SAS (62.5%, p < 0.05) or HSC-3 (58.1%, p < 0.05) cells, compared to their wild-type counterparts; interestingly the repressive effect of shFAT1 on the proliferation/viability of normal gingival fibroblast cell line HGF, was barely apparent and statistically insignificant (Figure 4A), indicating the non-lethality of shFAT1 to normal oral cavity cells, and at least in part, the oncoselectivity of the loss-of-FAT1 function in patients with OSCC. Using Ki-67 protein immunofluorescence staining, we demonstrated further that shFAT1 induced 80.6% (p < 0.01) and 78.3% (p < 0.01) reduction in the number of viable SAS and HSC-3 cells, respective, while conversely having no effect on the HGF cells which increased by 6% (Figure 4B). In parallel assays, we demonstrated that shFAT1 significantly down-regulated the expression levels of the proliferating cell nuclear antigen (PCNA) and B cell lymphoma 2 (Bcl2) proteins, while up-regulating Bax protein expression in the transfected SAS and HSC-3 cells, compared to their wild-type counterparts (Figure 4C), which is suggestive of a role for FAT1 in the promoting OSCC cell proliferation and evasion of cell death. Furthermore, gene expression-based heatmap generated from re-analysis of the A-AFFY-44, AFFY_HG_U133_PLUS_2, E-GEOD-30784 dataset of the gene expression profile of OSCC cohort (n = 229 samples, 54675 genes), originally to identify potential biomarkers for early detection of invasive OSCC using OSCC (n = 167), oral dysplasia (n = 17) and normal oral  (n = 45) samples, demonstrated that FAT1 expression was marginal in the non-cancerous control samples, but high FAT1 expression was associated with high PCNA, BIRC5/Survivin, BCL2, and BCL2L1, while low FAT1 expression correlated with high FADD, BAX, PARP1, CASP3, CASP8, and CASP9 in the cancer tissues (Figure 4D) suggesting an association between reduced FAT1 expression and induction of cell death. For a balanced perspective, we also observed that BAX was high in some of the FAT1high samples, however the median BAX/BCL2 ratio in the FAT1high samples was ~2.83-fold higher than in the FAT1low samples (Figure 4D). Singular value decomposition (SVD)-calculated principal component analysis (PCA) of same E-GEOD-30784 OSCC cohort further shows that while most (95.8%) OSCC sample were FAT1high, 4 were FAT1low (2.4%), and 3 were ambiguous (Figure 4E). These results indicate that the loss-of-FAT1 function significantly impairs the proliferation of OSCC cells and enhances apoptosis, but spares the normal or non-malignant cells of the oral cavity.

Q1.7.    Knockdown endogenous FAT1 could cause cell cycle arrest or cell death in OSCC cell lines? In figure 4B, Ki67, FAT1 and DAPI (nucleus stain) should be presented in the same panel. The endogenous FAT1 protein expression level should be showed in figure 4C.

A1.7.    We thank the reviewer for this comment. We have now included the DAPI nuclear staining as requested by the reviewer. We have also included data for Ki67 in Figure 4C, we have added the Ki67 data in our revised manuscript. Please kindly see our revised Results section, Pages 9-10, Lines 364-378:

Using Ki-67 protein immunofluorescence staining, we demonstrated further that shFAT1 induced 80.6% (p < 0.01) and 78.3% (p < 0.01) reduction in the number of viable SAS and HSC-3 cells, respective, while conversely having no effect on the HGF cells which increased by 6% (Figure 4B). In parallel assays, we demonstrated that shFAT1 significantly down-regulated the expression levels of marker of proliferation Ki-67, the proliferating cell nuclear antigen (PCNA) and B cell lymphoma 2 (Bcl2) proteins, while up-regulating Bax protein expression in the transfected SAS and HSC-3 cells, compared to their wild-type counterparts (Figure 4C), which is suggestive of a role for FAT1 in the promoting OSCC cell proliferation and evasion of cell death.

Also kindly see our updated Figures 4B and 4C and its legend, Page 10, Lines 393-415:

Figure 4. Loss-of-FAT1 function impairs OSCC cell proliferation and enhances cell death. (A) Images (upper panel) and graphical representation (lower panel) of the effect of shFAT1 on the viability/proliferation of SAS, HSC-3 or HGF cells. (B) Representative Ki-67 protein immunofluorescence staining images in wild-type and shFAT1 SAS, HSC-3 or HGF cells. DAPI serve as nuclear marker. (C) Representative western blot images of the differential expression of Ki-67, PCNA, Bcl2, and Bax proteins in wild-type or shFAT1 SAS and HSC-3 cells.  (D) Heatmap of FAT1, PCNA, BIRC5, BCL2, BCL2L1, FADD, BAX, PARP1, CASP3, CASP8, and CASP9 gene expression profile of OSCC cohort (n=229 samples, 54675 genes) using the A-AFFY-44, AFFY_HG_U133_PLUS_2, E-GEOD-30784 dataset. Rows are centered; unit variance scaling is applied to rows. Both rows and columns are clustered using correlation distance and average linkage. 23 rows, 229 columns. (E) Principal component analysis of same E-GEOD-30784 OSCC cohort. Unit variance scaling is applied to rows; SVD with imputation is used to calculate principal components. X and Y axis show principal component 1 and principal component 2 that explain 25.4% and 14.3% of the total variance, respectively. Prediction ellipses are such that with probability 0.95, a new observation from the same group will fall inside the ellipse. N = 229 data points. b-actin served as loading control. *p<0.05, **p<0.01, ***p<0.001.

Q1.8.    Does FAT1 expression can be a prognostic factor in OSCC patients clinically?

A1.8.    We are grateful for the reviewer’s question. This is indeed the goal of this present work, and based on the preclinical evidence provided herein, our answer to the reviewer’s question is in the affirmative. In fact parallel studies involving survival analyses in murine OSCC tumor xenograft models, and making use of a larger OSCC cohort to evaluate the correlation between FAT1, patients’ clinicopathological characteristics, performance status, and prognosis is currently ongoing and the results will be made available to the public as soon as available.

Having said that, proof-of-principle evidence provided in this present study indicate that FAT1 expression may be a prognostic factor in patients with OSCC.  Please kindly see our revised Results section, Page 6, Lines 224-239:

3.1. High FAT1 expression drives HNSC, defines high risk population and is associated with poor prognosis.

In the light of the divergent roles of FAT1 in different malignancies, seeking to understand the pathocytological relevance of FAT1 and determine its molecular dynamics in highly metastatic and recurrent OSCC cells, we examined FAT1 expression and mutational profile in the TCGA HNSC cohort (n = 502). Results of our bioinformatics analysis showed that of the 131 mutated/mutant cancer drivers detected in the TCGA HNSC cohort, the most mutated drivers included TP53, FAT1, NOTCH1, CDKN1, CDKN2A and PIK3CA, in decreasing order of mutational frequency (Figure 1A). In addition, we observed that of the principal 97 mutations in FAT1, the truncating (otherwise known as nonsense) gene mutation accounting for 75% of all FAT1-associated mutations, was most frequent (Figure 1B). Because of the implication of driver mutation burden in poor clinical outcome (22), we evaluated the effect of altered FAT1 expression in the TCGA HNSC cohort, and demonstrated that compared to the FAT1low group, high FAT1 (FAT1high) expression confers significant survival disadvantage on the patients with HNSC ((Hazard ratio (HR) = 1.34, 95% CI: 1.02 - 1.77; p = 0.038) (Figure 1C). Consistent with the above, the FAT1high group exhibited strong association with high risk of disease-specific mortality compared with the FAT1low group (Figure 1D), suggesting a prognostic role for FAT1 in patients with HNSC.

Please also kindly see our updated Figures 1C and 1D, with their legends, Pages 6-7, Lines 239-248:

Figure 1. High FAT1 expression drives HNSC, defines high risk population and is associated with poor survival. (A) Driver cloud showing the most recurrently mutated cancer driver genes in the TCGA-HNSC dataset; The gene symbol size corresponds to count of samples with protein-affecting mutations (PAMs). (B) Mutations distribution along FAT1 protein sequence. (C) KM plot of the FAT1 expression-stratified overall survival of the TCGA HNSC June 2016 cohort (n =502). (D) Correlative functional analysis of FAT1 expression, risk, prognosis and survival in the TCGA-HNSC cohort.

Also see our revised Results section, Page 7, Lines 249-270:

3.2. High FAT1 expression is positively correlated with disease progression and poor clinical outcome in patients with HNSC.

In the light of our data suggesting a driver and prognostic role for FAT1 mutations and expression, respectively, in patients with HNSC, probing further for validation of these function, we observed a strong association between increased expression of FAT1 (probe ID: 201579_at) and disease progression, as demonstrated by mean expression transformed counts of 2.31, 1.07, 1.2, and 1.46 in stage I (T1), stage II (T2), stage III (T3) and stage IV (T4), compared with 0.97 in normal samples (Figure 2A), based on analysis of the GSE3524/GDS1584 dataset of gene expression profiles in OSCC cells from 16 patients with 20 samples, isolated by laser capture microdissection. Interestingly, FAT1 expression spikes in this dataset were markedly associated with nodal involvement as seen in T1N2bM0, T3N2bM0, T4N1M0, and T4N2bM0 (Figure 2A, also see Suppl. Figure S4). Our analysis of the OSCC cohort (n = 291) of the TCGA HNSC dataset showed that aberrant expression of FAT1 in patients with OSCC was strongly associated with worse overall survival in the FAT1high, compared to the FAT1low patients [HR: 1.12 (0.97 - 1.3), p = 0.121] with the median survival for FAT1high and FAT1low patients being 1081 and 1591 days, respectively (Figure 2B). Moreover, analysis of the E-MTAB-1328 dataset from methylome, transcriptome and miRNome array and high throughput sequencing profiling of 89 patients with 104 samples also showed that FAT1high patients exhibited shorter metastasis-free survival than their FAT1low counterparts [HR:1.1 (0.74 - 1.63), p = 0.653] (Figure 2C) and shorter relapse-free survival [HR:2.35 (0.48 - 11.52), p = 0.290] in the FAT1high, compared to the FAT1low patients, based on analysis of 44 primary head and neck tumor samples using the GSE10300 dataset (Figure 2D). These data are indicative of a critical role of high FAT1 expression in disease progression and poor clinical outcome in patients with HNSC.

Kindly also see our Figure 2, and its legend, Page 7, Lines 272-280:

Figure 2. High FAT1 expression is positively correlated with disease progression and poor clinical outcome in patients with HNSC. (A) Histograms of the stage-associated differential expression of FAT1 in the GPL96/GDS1584 gene expression profiling by array data-set, transformed count, 20 samples. Kaplan-Meier plots of the correlation between differential FAT1 expression and (B) overall survival of OSCC cohort of the TCGA - HNSC, n = 291, (C) metastasis-free survival of patients with HNSC, n = 89 using the E-MTAB-1328 dataset, and (D) relapse-free survival of patients  undergoing surgery or biopsy for HNSC at University of North Carolina, Chapel Hill and Vanderbilt University using the GSE10300 dataset, n = 44. High/Low expression bifurcation was based on median FAT1 expression in all cases.

Please also see our newly included Supplementary Figure S4 and its legend, Pages 20-21, Lines 226-234:

Supplementary Figure S4. High FAT1 expression is positively correlated with nodal involvement and poor clinical outcome in patients with HNSC.  Boxplot and whiskers plots of the correlation between (A) clinical node (N) stage, or (B) number of lymph-nodes positive by IHC, and the differential expression of FAT1 in TCGA-HNSC cohort. N, nodal; IHC, immunohistochemistry. Kaplan-Meier plots of the correlation between differential FAT1 expression and (C) relapse-free survival in the Chung Parker Head and Neck GSE10300  cohort, n = 44, or  (D) overall survival in the TCGA-HNSC June 2016 cohort, n = 502.

Q1.9.   Is FAT1 expression in OSCC patients associated with clinicopathologic parameters such as AJCC stage, T classification, N classification, tumor grade, perineural invasion, lymphovascular invasion, extranodal extension…etc?

A1.9.    We are grateful for the reviewer’s insightful question. As with the reviewer, the goal of all our work is always clinical applicability. Thus, as already indicated above, parallel studies making use of a large OSCC cohort to evaluate possible correlations between FAT1 expression, patients’ clinicopathological characteristics, performance status, and prognosis is currently ongoing and the results will be made available to the public as soon as available.

Q1.10. FAT1 knockdown could cause tumor growth decrease in vivo?

A1.10. We are grateful for the reviewer’s insightful question. We agree with the reviewer that such data would be convincing and help our case. While it may be inferred from preclinical data provided herein that FAT1 knockdown could cause decrease of tumor growth in vivo, parallel studies involving survival analyses in murine OSCC tumor xenograft models, and making use of a larger OSCC cohort to evaluate the correlation between FAT1, patients’ clinicopathological characteristics, performance status, and prognosis is currently ongoing and the results will be made available to the public as soon as available.

Q1.11. This manuscript should be English editing.

A1.11. We are sincerely thank the reviewer for this comment. We have now carefully reviewed our manuscript for likely English language grammatical, orthographic and syntactic errors. Please kindly see our revised manuscript.

Q1.12. The experimental protocol of figure 5A and B needed to describe in the materials and methods.

A1.12. We thank the reviewer for the keen observation and for pointing out this omission to us. As requested by the reviewer, experimental protocol for Figures 5A and 5B (i.e. FAT1 knockdown, migration, and invasion assays) can now be found in our revised manuscript.  Please kindly see our revised Materials and Methods section, Page 4, Lines 151-159:

2.6. shFAT1 transfection and establishment of stable knockdown cell lines

The small hairpin RNA (shRNA) specifically targeting human FAT1 (FAT1 shRNA Plasmid (h); #sc-88872-SH) obtained from Santa Cruz Biotechnology Inc. (Santa Cruz, CA, USA ) was transfected into HSC3 or SAS cells grown in 6-well plates to 60% confluence using Lipofectamine 2000 reagent (Invitrogen) according to the manufacturer’s instructions. For puromycin selection of stably transfected clones, 48 h after shFAT1 transfection, medium was aspirated and replaced with fresh medium containing 5 mg/mL puromycin and incubated for another 48 h in humidified 5% CO2 atmosphere incubator at 37 °C. Thereafter, shFAT1-transfected cells were harvested for qRT-PCR or western blot analysis.

Please kindly see our revised Materials and Methods section, Page 5, Lines 162-177:

2.7. Cell migration and invasion assays

Cell migration capability was evaluated using the wound healing assay. Briefly, wild-type or shFAT1-transfected HSC-3 or SAS cells were seeded into 6-well plates (Corning Inc., Corning, NY, USA) containing complete growth media supplemented with 10% FBS, and cultured to ≥ 98% monolayer confluency. The cell monolayers were scratched with sterile yellow pipette tips to denude the culture wells. Images of cell migration were captured at the 0 and 24 h time-points after denudation, under a microscope with a 10× objective lens, and later  analyzed with the NIH ImageJ software (https://imagej.nih.gov/ij/download.html).

For invasion assay, using 24-well plate matrigel Transwell® systems, we seeded 3 × 104 cells into the upper chamber of the insert (BD Bioscience, pore size = 8 μm) containing FBS-free media, while the lower chamber contained 10% FBS-supplemented media. After incubation for 24 h, all media were carefully discarded, non-invaded cells in the upper surface of the insert were removed carefully with sterile cotton swipes, while invaded cells on the under side of the membrane were stained with crystal violet dye after fixture with 3.7% formaldehyde, and then the average number of invaded cells were determined under microscope, from at least five non-overlapping visual fields selected randomly.

Q1.13. In introduction, authors described” ………definitive chemoradiotherapy (CRT) for patients with advanced-stage tumors, where chemotherapy consists of cisplatin (CDDP), docetaxel (DTX), and 5-flurouracil (5-FU), especially for patients with advanced-stage malignancies, however, this is often associated with increased risk of severe therapy-related toxicities and adverse effects, including osteoradionecrosis (ORN),…”. Osteoradionecrosis is the long-term side effect of radiotherapy, not the side effect of chemotherapy. It is better if authors can mention the side effect of chemotherapy such as neutropenia

A1.13. We really thank the reviewer for this suggestion. We have now included neutropenia as suggested by the erudite reviewer. Please kindly see our revised Introduction section, Page 2, Lines 62-69:

Currently, the standard of care remains surgery for patients with early (I and II) - or advanced (III and IV) - stage tumors, and definitive chemoradiotherapy (CRT) for patients with advanced-stage tumors, where chemotherapy consists of cisplatin (CDDP), docetaxel (DTX), and 5-flurouracil (5-FU), especially for patients with advanced-stage malignancies (5, 6), however, this is often associated with increased risk of severe therapy-related toxicities and adverse effects, including neutropenia and osteoradionecrosis (ORN), increasing incidence of therapy failure and disease relapse, and low median survival rates for patients with OSCC (3), as indicated by a 5-year survival rate that has remained consistently below 50% over the last 3 decades (7).

Q1.14. In introduction, authors described “…Despite the association of adjuvant chemotherapy with enhanced survival of patients with advanced stage OSCC…”. According to the randomized large study, adjuvant chemoradiotherapy can improved survival of advanced OSCC after operation. However, it is still controversial for adjuvant chemotherapy.

A1.14. We sincerely appreciate the reviewer’s comment and the insight conveyed therein. In the particular instance referred to by the reviewer, we alluded to ‘adjuvant chemotherapy’ in the context of its definition as ‘a therapeutic approach that combines with or accompanies another, to enhance the effect of the later, and lower the risk of disease recurrence after treatment’; thus in the reference (ref.8) cited, adjuvant chemotherapy was addressed, being consistent with our study on FAT1 and cisplatin-resistance. Please kindly see our revised Introduction section, Page 3, Lines 70-79:

Despite the association of adjuvant chemotherapy with enhanced survival of patients with advanced stage OSCC, and the touted therapeutic promise of Cisplatin (cis-diamminedichloroplatinum, CDDP), an alkylating chemotherapeutic agent, with well-documented anticancer efficacy for patients with cancers, including OSCC, intrinsic insensitivity or acquired resistance to cisplatin and disease recurrence continues to impede treatment success and plague survival of patients with OSCC (8). Thus, the need to unravel probable mechanisms underlying cisplatin-resistance, which invariably will enable the the discovery of novel actionable molecular or therapeutic targets and development of more effective treatment strategies to abrogate chemoresistance and improve the clinical outcome of patients with OSCC.

We sincerely wish to thank the reviewer for taking time off his busy schedule to review our work and provide several constructive critiques and valuable suggestions. We have made use of all these suggestions and believe that they have helped improve the scientific quality and appeal of our work. We are indeed grateful.

Reviewer 2 Report

The manuscript entitled “Targeting FAT1 inhibits carcinogenesis, induces oxidative stress and enhances cisplatin sensitivity through deregulation of LRP5/WNT2/GSS signaling axis in Oral Squamous Cell Carcinoma” is an interesting work evaluating FAT atypical cadherin 1 in oral carcinogenesis. In particular, this work showed that FAT1 is a driver oncogene aberrantly expressed in OSCC and related to the metastatic phenotype.

The Authors demonstrated that silencing FAT1 attenuated migration, invasion, and clonogenicity of OSCC cells and sensitized cisplatin-resistant OSCC cells to chemotherapy. The techniques used were appropriate and detailed enough. This is a well-designed study with rigorous methods. The discussion is well-balanced and the statements are supported by the data. The presence of a graphical abstract and a list of research highlights helps the reader even more quickly to understand.

I suggest some minor revision to improve the paper:

Delete some acronyms (e.g. CRT, DTX, 5-FU, ORN) since they are used only one time in the text. As the importance of the topic, I suggest adding some considerations related to the epidemiology and prognosis of OSCC. In particular, the unchanging survival in patients with OSCC underscores the need for better prognostic tools, as recently reported in the 8th edition of American Joint Committee on Cancer staging system [1]. Few minor language corrections should be necessary (e.g. the sentence written in page 9 lines 294 “As with all…problems,” in unnecessary).

[1] Mascitti M, et al. American Joint Committee on Cancer staging system 7th edition versus 8th edition: any improvement for patients with squamous cell carcinoma of the tongue? Oral Surg Oral Med Oral Pathol Oral Radiol. 2018 Nov;126(5):415-23

Author Response

Response to Reviewers:

Point-by-point responses to reviewer’s comments:

We would like to thank the reviewer for the thorough reading of our manuscript as well as their valuable comments. We believe all comments are borne out of good faith, and thus, have tried to address their comments conscientiously and feel that they have further improved the readability and appeal of our work, as well as strengthened the manuscript. Below are our point-by-point responses.

Q2.1.    In this study, The manuscript entitled “Targeting FAT1 inhibits carcinogenesis, induces oxidative stress and enhances cisplatin sensitivity through deregulation of LRP5/WNT2/GSS signaling axis in Oral Squamous Cell Carcinoma” is an interesting work evaluating FAT atypical cadherin 1 in oral carcinogenesis. In particular, this work showed that FAT1 is a driver oncogene aberrantly expressed in OSCC and related to the metastatic phenotype.

The Authors demonstrated that silencing FAT1 attenuated migration, invasion, and clonogenicity of OSCC cells and sensitized cisplatin-resistant OSCC cells to chemotherapy. The techniques used were appropriate and detailed enough. This is a well-designed study with rigorous methods. The discussion is well-balanced and the statements are supported by the data. The presence of a graphical abstract and a list of research highlights helps the reader even more quickly to understand.

I suggest some minor revision to improve the paper:

A2.1.    We are very grateful to the reviewer for taking time to read our manuscript and for the critiques and suggestions made in order to help us improve the quality of our work. We are even more thankful for the words of encouragement by the erudite reviewer. In this revised manuscript, we have made effort to make address all the comments and suggestions.

Q2.2.    In this study, Delete some acronyms (e.g. CRT, DTX, 5-FU, ORN) since they are used only one time in the text. As the importance of the topic, I suggest adding some considerations related to the epidemiology and prognosis of OSCC. In particular, the unchanging survival in patients with OSCC underscores the need for better prognostic tools, as recently reported in the 8th edition of American Joint Committee on Cancer staging system [1]. Few minor language corrections should be necessary (e.g. the sentence written in page 9 lines 294 “As with all…problems,” in unnecessary).

[1]Mascitti M, et al. American Joint Committee on Cancer staging system 7th edition versus 8th edition: any improvement for patients with squamous cell carcinoma of the tongue? Oral Surg Oral Med Oral Pathol Oral Radiol. 2018 Nov;126(5):415-23

A2.2.    We appreciate the reviewer’s comments. As suggested by the reviewer, we have now deleted the said acronyms in our revised manuscript. Please kindly see our revised Introduction section, Page 2, Lines 62-69:

Currently, the standard of care remains surgery for patients with early (I and II) - or advanced (III and IV) - stage tumors, and definitive chemoradiotherapy for patients with advanced-stage tumors, where chemotherapy consists of cisplatin (CDDP), docetaxel, and 5-flurouracil, especially for patients with advanced-stage malignancies (5, 6), however, this is often associated with increased risk of severe therapy-related toxicities and adverse effects, including neutropenia and osteoradionecrosis, increasing incidence of therapy failure and disease relapse, and low median survival rates for patients with OSCC (3), as indicated by a 5-year survival rate that has remained consistently below 50% over the last 3 decades (7).

Regarding “the unchanging survival in patients with OSCC…the need for better prognostic tools”, we find the suggested text to be a perfect fit and have made use of the reviewer’s suggested text. Please kindly see our revised Introduction section, Page 3, Lines 74-85:

Despite the association of adjuvant chemotherapy with enhanced survival of patients with advanced stage OSCC, and the touted therapeutic promise of Cisplatin (cis-diamminedichloroplatinum, CDDP), an alkylating chemotherapeutic agent, with well-documented anticancer efficacy for patients with cancers, including OSCC, intrinsic insensitivity or acquired resistance to cisplatin and disease recurrence continues to impede treatment success and plague survival of patients with OSCC (8). Thus, the need to unravel probable mechanisms underlying cisplatin-resistance, which invariably will enable the the discovery of novel actionable molecular or therapeutic targets and development of more effective treatment strategies to abrogate chemoresistance and improve the clinical outcome of patients with OSCC. Recent work by Mascitti M, et al, very well encapsulates the epidemiological implication and prognostic ramifications of this dearth of therapeutically-relevant actionable targets and prognostigators, and further begs the case for same (9).

Also kindly see our updated Reference section, Page 18, Lines 700-703:

Mascitti M, Rubini C, De Michele F, Balercia, Girotto R, Troiano G, Muzio LL, Santarelli.. American Joint Committee on Cancer staging system 7th edition versus 8th edition: any improvement for patients with squamous cell carcinoma of the tongue? Oral Surg Oral Med Oral Pathol Oral Radiol. 2018; 126(5):415-23 doi:10.1016/j.oooo.2018.07.052 

Addressing “Few minor language corrections should be necessary (e.g. the sentence written in page 9 lines 294 “As with all…problems,” in unnecessary)”. We have deleted the phrase in question in our revised manuscript. Please kindly see our revised Results section, Page 9, Lines 351-358:

3.4. Loss-of-FAT1 function impairs OSCC cell proliferation and enhances cell death.

To proffer solution, we investigated if and to what degree the loss-of-FAT1 function negatively affects the survival and proliferation of OSCC cells.  Our results indicate that shFAT1 transfection markedly suppressed the proliferation cum viability of SAS (62.5%, p < 0.05) or HSC-3 (58.1%, p < 0.05) cells, compared to their wild-type counterparts; interestingly the repressive effect of shFAT1 on the proliferation/viability of normal gingival fibroblast cell line HGF, was barely apparent and statistically insignificant (Figure 4A), indicating the non-lethality of shFAT1 to normal oral cavity cells, and at least in part, the oncoselectivity of the loss-of-FAT1 function in patients with OSCC.

We have now carefully reviewed our manuscript for likely English language grammatical, orthographic and syntactic errors. Please kindly see our revised manuscript.

We sincerely wish to thank the reviewer for taking time off his busy schedule to review our work and provide several constructive critiques and valuable suggestions. We have made use of all these suggestions and believe that they have helped improve the scientific quality and appeal of our work. We are indeed grateful.

Reviewer 3 Report

The present study assessed the probable implication of FAT1 in the oncogenicity, metastatic and therapy-resistance phenotypes of OSCC cells and the poor prognosis of patients with OSCC. The authors demonstrated that FAT1 enhanced cancer cell proliferation, chemoresistance, impaired cisplatin-induced cell death, and that the therapeutic targeting of FAT1 re-sensitized cisplatin-resistant OSCC cells to cisplatin, thus projecting FAT1 as a novel therapeutic target for anticancer treatment of therapy-resistant OSCC. While the topic itself is interesting and falls within the journal’s scope, the main shortcoming of the study is the limited number of OSCC patient samples (n = 21). OSCC is relatively common, and thus (n) in my opinion is too low to justify the findings, in addition to the sparse information of the selected patients. I recommend authors to increase the number of tested OSCC cases (and, naturally, to add/match the healthy controls appropriately). Were there any sample size calculations? I think this is important in such context. In the background, the reference cited to indicate the incidence of OSCC is rather old. I think this should be updated. Please use a newer reference, e.g. GLOBOCAN 2018/or JAMA report 2019. FAT1 has been extensively studied in OSCC and has been linked to potential therapeutic approaches. Authors should carefully state, in the introduction and discussion, what does their study add to the previous published literatures (e.g., NIE et al., 2016; Lin et al., 2018). Indeed, gene expression and phenotype can vary between low passage and high passage cell lines. Consequently, higher passages cell lines no longer represent reliable models of the original source tissue (e.g. OSCC). Please provide full details of the used cell lines including passage number and why such concentrations were selected.

Author Response

Response to Reviewers:

Point-by-point responses to reviewer’s comments:

We would like to thank the reviewer for the thorough reading of our manuscript as well as their valuable comments. We believe all comments are borne out of good faith, and thus, have tried to address their comments conscientiously and feel that they have further improved the readability and appeal of our work, as well as strengthened the manuscript. Below are our point-by-point responses.

Q3.1.    The present study assessed the probable implication of FAT1 in the oncogenicity, metastatic and therapy-resistance phenotypes of OSCC cells and the poor prognosis of patients with OSCC. The authors demonstrated that FAT1 enhanced cancer cell proliferation, chemoresistance, impaired cisplatin-induced cell death, and that the therapeutic targeting of FAT1 re-sensitized cisplatin-resistant OSCC cells to cisplatin, thus projecting FAT1 as a novel therapeutic target for anticancer treatment of therapy-resistant OSCC.

A3.1.    We are very grateful to the reviewer for taking time to read our manuscript and for the critiques and suggestions made in order to help us improve the quality of our work. In this revised manuscript, we have made effort to make address all the comments and suggestions.

Q3.2.    While the topic itself is interesting and falls within the journal’s scope, the main shortcoming of the study is the limited number of OSCC patient samples (n = 21). OSCC is relatively common, and thus (n) in my opinion is too low to justify the findings, in addition to the sparse information of the selected patients. I recommend authors to increase the number of tested OSCC cases (and, naturally, to add/match the healthy controls appropriately).

A3.2.    We sincerely appreciate the reviewer’s comment. We agree with the erudite reviewer that the main shortcoming of the study is the limited number of OSCC patient samples (n = 21), considering the relative prevalence of OSCC in Asia. However, this is indeed a ‘Proof-of-principle’ study, and accordingly since the goal of our work is always clinical applicability, there are parallel studies making use of a larger OSCC cohort to evaluate possible correlations between FAT1 expression, patients’ clinicopathological characteristics, performance status, therapeutic response, and prognosis is currently ongoing and the results will be made available to the public as soon as available.

Q3.3.    Were there any sample size calculations? I think this is important in such context

A3.3.    We sincerely appreciate the reviewer’s question. The number of samples used in the present ‘proof-of-principle’ study was based on the following calculations using online software for determination of sample size:

Confidence level: 95% (selected since most research work use the 95% confidence level; meaning we were 95% certain of results obtained)

Confidence Interval: 4 and 47%

Population: 447 751 (based n an annual incidence of 447, 751 cases (ref (1) )

Sample size needed: 5 (automatically computed)

Based on past experience and our understanding that when determining the sample size needed for a given level of accuracy it is important to use the worst case percentage, which is 50%, especially in a case like ours where we wanted to determine a general level of accuracy for a sample you already have.

Thus, we used a confidence interval of 4 and 47%; where using a confidence interval of 4 and 47% meant we can be "sure" that our model in the worst case scenario would be representative of the population between 43% (47-4) and 51% (47+4).

Thus, together, putting the confidence level and the confidence interval together, we could say that we are 95% sure that the true percentage of the population is between 43% and 51%.

In this study, we have provided 21 pairs of non-tumor/OSCC samples, which is more than the required sample size needed (n = 5).

Q3.4.    In the background, the reference cited to indicate the incidence of OSCC is rather old. I think this should be updated. Please use a newer reference, e.g. GLOBOCAN 2018/or JAMA report 2019.

A3.4.    We are very grateful for the reviewer’s comment. As requested by the reviewer, we have now updated the incidence data in our revised manuscript. Please kindly see our revised Introduction section, Page 2, Lines 52-63:

Introduction

Oral cancer, comprising cancers of the oral cavity (International Statistical Classification of Diseases and Related Health Problems, ICD-10: C00-06) and oropharyngeal regions (ICD-10: C09-10), is one of the most prevalent malignancies in the world, with an annual incidence of 447, 751 cases (1), and with a predicted 67.1% rise in the disease-specific mortality by 2035 (n = 242, 886) compared with 2012 (n = 145, 353), oral cancer remains a principal cause of cancer-related mortality globally (1, 2). Though multifactorial, the commonest risk factors associated with oral cancer include gene susceptibility, betel nut chewing, tobacco smoking, alcohol abuse, and human papillomavirus (HPV) infection (3, 4). The oral squamous cell carcinoma (OSCC) histological type constitutes more than 90% of all oral cancer, and is highly invasive, most often insensitive to chemo- and/or radiation therapy, and associated with high incidence of recurrence, and poor survival rates (1, 2).

Please also kindly see our updated Reference section, Page 18, Lines 685-687:

Ferlay J, Ervik M, Lam F, Colombet M, Mery L, Piñeros M, Znaor A, Soerjomataram I, Bray F (2018). Global Cancer Observatory: Cancer Today. Lyon, France: International Agency for Research on Cancer. Available from: https://gco.iarc.fr/today, accessed [08 November 2019].

Q3.5.    FAT1 has been extensively studied in OSCC and has been linked to potential therapeutic approaches. Authors should carefully state, in the introduction and discussion, what does their study add to the previous published literatures (e.g., NIE et al., 2016; Lin et al., 2018).

A3.5.    We appreciate the reviewer’s comment. As requested by the reviewer, we have carefully stated what our present study adds to the contemporary theme-relevant knowledge in our revised manuscript. Please kindly see our revised Introduction section, Page 3, Lines 90-117:

The last two decades has been characterized by increased interest in and accumulating evidence of the critical role of the tumor microenviroment (TME) in the initiation and progression of cancer, as well as the response of cancerous cells to therapeutic modality (10, 11, 12). Cell adhesion molecules, including cadherins are vital components of the TME, being associated with cell-cell adhesive bonds in solid tissues, immune modulation, promotion of cancer growth, metastasis and survival (13)  The role of atypical cadherins such as the FATs in tumor formation, dissemination, and prognosis, has garnered attention lately. The human FAT gene family, consisting of FAT1, FAT2, FAT3 and FAT4 genes, encodes large proteins with extracellular Cadherin repeats, EGF-like domains, an Laminin-G-like domains, where human FAT1, FAT2, and FAT3 are orthologous with Drosophila Fatl, and human FAT4 authologous with Drosophila Fat (14). While there is ample information on the role(s) of the FAT atypical cadherin 1 (FAT1) in contemporary literature, these information are rather conflicting, with FAT1 suggested as a tumor suppressor based on its inhibition of Yes-associated protein (YAP)1 function and suppression of cell growth (15), inhibition of epithelial-to-mesenchymal transition (EMT) in esophageal squamous cell cancer (16), suppression of the invasive capability, nodal involvement, lymphovascular permeation and tumor recurrence in HNSCC (17), and its loss-of-function eliciting resistance to cyclin dependent kinase (CDK)4/6 inhibitors in ER+ breast cancer (18). Conversely, aberrant expression of FAT1 has been implicated in the high invasiveness of GBM cells (19), cancerous cell proliferation, apoptosis evasion and disease progression in hepatocellular carcinoma (HCC) (20), relapse and poor prognosis in patients with B-cell acute lymphoblastic leukemia (21).

Against the background of this ambivalent context-dependent role of FAT1 in malignancies and its under-explored role in OSCC, the present study investigated the probable implication of FAT1 in the oncogenicity, metastatic and therapy-resistance phenotypes of OSCC cells and the poor prognosis of patients with OSCC. Herein, we demonstrated that aberrantly expressed FAT1 by cancerous cells enhanced their proliferation, promoted chemoresistance, impaired cisplatin-induced cell death, and that the therapeutic targeting of FAT1 re-sensitized cisplatin-resistant OSCC cells to cisplatin through deregulation of LRP/Wnt signaling, thus projecting FAT1 as a novel therapeutic target for anticancer treatment of therapy-resistant OSCC.

Also kindly see our revised Discussion section, Pages 16-17, Lines 581-633:

Regardless of the aforementioned consistencies between our present findings and several other published works, we do acknowledge our findings contradict those of Lin SC et al. (17), suggesting FAT1 acts as a tumor suppressor, with lower FAT1 protein expression bearing significant correlation with lymph node metastasis, lymphovascular permeation, tumor recurrence, and shorter disease-free survival (DFS) in patients with HNSC. We cannot fully explain this contradiction, however, we cautiously attribute this to tumor heterogeneity and/or cohort constitution, based on demonstrated association of high expression of FAT1 with high grade cancerous cells and low FAT1 with low grade cancerous cells in previous works (19, 30), as observed in the present work with the use of the HSC-3 and SAS cells which are poorly differentiated human squamous carcinoma of the tongue cells with high lymph node metastasis potential (34 - 36), as well as with our use of OSCC cohorts with more high grade, advanced stage or metastatic cases. As rightly noted by Soussi T and Wiman KG for p53 (TP53) (37), it is not impossible that while the standard criteria for definition of various cancer genes may confine the tumor protein FAT1 to the role of a tumor suppressor, accruing evidence across multiple cancer types with diverse histology, indicate that FAT1 does indeed act as an oncogene.

The present study thus provides rationale to look outside the box of classical and traditional classification of FAT1 as a tumor suppressor, by highlighting various oncogenic properties that make FAT1 a putative therapeutic target that should not be underestimated in OSCC. Consistent with rationalizations on genes with both oncogenic and tumor-suppressor functions by Shen L, et al. (38), it is also probable that the function- altering mutations in FAT1 are the main driving force in the FAT1-facilitated oncogenesis and cisplatin resistance in OSCC documented herein. Interestingly, like TP53, the mode of FAT1 inactivation is quite unique, compared with most tumor suppressors; ~22% and 75% of FAT1 genomic alterations are missense and truncating/nonsense mutations, respectively, both of which facilitate the synthesis of a stable mutant FAT1 protein which accumulates in the plasma membrane and nucleus of the aggressive and/or cisplatin-resistant OSCC cells. Comparatively, this high frequency of amino acid substitution (missense) or premature termination of translation (truncating/nonsense) is highly analogous with various cancer types, regardless of the difference in mutation spectrum (37). Finally, consistent with Muller’s exposition on the nature and causes of gene mutations, based on the classification of mutations hinged on genophenotypic analyses (39), genomic alterations in FAT1 are akin to the ‘amorph’ or ‘hypomorphic’ mutation which are more characteristic of so-called ‘tumor suppressors’, wherein the tumor suppressor function is totally impaired or a partial reduced, resulting in its acquisition of oncogenicity and ability to drive cancer. While it is conceivable that missense and nonsense/truncating mutations resulting in true amorphic variants elicit complete loss of tumor suppressor function, in many instances it is difficult to exclude residual activities that result in heterogeneous hypomorphic variants with context-dependent functional duality as documented in the present study for the protocadherin FAT1. This rationalization based on tumor heterogeneity and mutational status are therapeutically relevant as they go beyond the initial biological function ascribed to FAT1, and can inform discovery or development of novel anti-OSCC therapeutic strategies with high efficacy.

Please also see our Research Highlights section, Page 1, Lines 23-28:

Research Highlights:

FAT1 is aberrantly expressed in human OSCC and influences survival rate. FAT1 is a driver oncogene and modulates the metastatic phenotype of OSCC cells. Silencing FAT1 attenuates migration, invasion, and clonogenicity of OSCC cells. Suppression of FAT1 enhances oxidative stress and re-sensitizes cisplatin-resistant OSCC cells to cisplatin therapy.

Q3.6.    Indeed, gene expression and phenotype can vary between low passage and high passage cell lines. Consequently, higher passages cell lines no longer represent reliable models of the original source tissue (e.g. OSCC). Please provide full details of the used cell lines including passage number and why such concentrations were selected

A3.6.    We are grateful for the reviewer’s comment. We are cognizant of the role of culture condition, including passage number on the cytogenetic and functional attributes of cells, and do agree with the reviewer. As requested by the reviewer, we have carefully stated what our present study adds to the contemporary theme-relevant knowledge in our revised manuscript. Please kindly see our revised Materials and Methods section, Page 4, Lines 148-159:

2.3. Cell lines and cell culture

Adult Human Primary Normal Gingival Fibroblast cells (HGF, ATCC® PCS-201-018™) obtained from the American Type Culture Collection, Manassas, VA, USA), was cultured in Fibroblast growth medium (#116-500, Sigma - Aldrich Inc., St. Louis, MO, USA). Human OSCC cell line with high metastatic potential HSC-3 (#JCRB0623) and SAS (#JCRB0260) were purchased from the National Institutes of Biomedical Innovation, Health and Nutrition (NIBIOHN)-Japanese Collection of Research Bioresources (JCRB) Cell Bank (Osaka, Japan), and cultured in Gibco™ Dulbecco’s Modified Eagle Medium (DMEM, #11995065, Thermo Fisher Scientific Inc., Bartlesville, OK, USA), supplemented with 10% fetal bovine serum (FBS, #26140079) and 1% penicillin-streptomycin at 37oC, in 5% humidified CO2 incubator. Cells used in the present study were all ≤ passage number 3 (P.3) and were sub-cultured when 90% confluent and media changed every 48h.

We sincerely wish to thank the reviewer for taking time off his busy schedule to review our work and provide several constructive critiques and valuable suggestions. We have made use of all these suggestions and believe that they have helped improve the scientific quality and appeal of our work. We are indeed grateful.

Reviewer 4 Report

The association between FAT1 mutation and overall survival in OSCC has been reported. Also, the effect of FAT1 knockdown and oncogenic phenotypes in HNSCC cells has been demonstrated in other studies.

Please avoid using long sentences to describe the conclusion. For instance, #551-562

What is the rationale to examine the expression of Bax and Bcl? Besides, it is suggested to briefly mention the rationale of measuring GSH. The connection between “FAT1 and apoptosis” or “FAT1 and oxidative stress” needs to be addressed.

The authors showed numerous factors that may be correlated with FAT1, it would be better to describe the reason to select the LRP/WNT signaling for further examination.

Author Response

We would like to thank the reviewer for the thorough reading of our manuscript as well as their valuable comments. We believe all comments are borne out of good faith, and thus, have tried to address their comments conscientiously and feel that they have further improved the readability and appeal of our work, as well as strengthened the manuscript. Below are our point-by-point responses.

Q4.1.    The association between FAT1 mutation and overall survival in OSCC has been reported. Also, the effect of FAT1 knockdown and oncogenic phenotypes in HNSCC cells has been demonstrated in other studies.

A4.1.    We are very grateful to the reviewer for taking time to read our manuscript and for the critiques and suggestions made in order to help us improve the quality of our work. In this revised manuscript, we have made effort to make address all the comments and suggestions.

We appreciate the reviewer’s comment. As alluded by the reviewer, we are cognizant of contemporary literature on the subject and have tried to cite many of these in the present work, while highlighting what the present work adds to current theme-relevant knowledge.

Q4.2.    Please avoid using long sentences to describe the conclusion. For instance, #551-562

A4.2.    We are grateful for the reviewer’s comment. We have now had our manuscript carefully reviewed by a native English-speaking colleague for likely English language grammatical, orthographic and syntactic errors. Please kindly see our revised manuscript.

Q4.3.    What is the rationale to examine the expression of Bax and Bcl? Besides, it is suggested to briefly mention the rationale of measuring GSH. The connection between “FAT1 and apoptosis” or “FAT1 and oxidative stress” needs to be addressed.

A4.3.    We sincerely thank the reviewer for these comments. The Bax/Bcl2 apoptosis index was determined to better understand probable underlying mechanism for reduced cell viability and proliferation in the OSCC SAS and HSC-3 cells, especially as apoptosis and cell proliferation are linked by cell-cycle regulators and apoptotic stimuli that affect both processes. Please kindly see our revised Results section, Pages 9-10, Lines 358-383:

3.4. Loss-of-FAT1 function impairs OSCC cell proliferation and enhances cell death.

To proffer solution, we investigated if and to what degree the loss-of-FAT1 function negatively affects the survival and proliferation of OSCC cells.  Our results indicate that shFAT1 transfection markedly suppressed the proliferation cum viability of SAS (62.5%, p < 0.05) or HSC-3 (58.1%, p < 0.05) cells, compared to their wild-type counterparts; interestingly the repressive effect of shFAT1 on the proliferation/viability of normal gingival fibroblast cell line HGF, was barely apparent and statistically insignificant (Figure 4A), indicating the non-lethality of shFAT1 to normal oral cavity cells, and at least in part, the oncoselectivity of the loss-of-FAT1 function in patients with OSCC. Using Ki-67 protein immunofluorescence staining, we demonstrated further that shFAT1 induced 80.6% (p < 0.01) and 78.3% (p < 0.01) reduction in the number of viable SAS and HSC-3 cells, respective, while conversely having no effect on the HGF cells which increased by 6% (Figure 4B). In parallel assays, against the background of cell-cycle regulated and apoptosis stimuli-mediated existent interconnection between cell proliferation and apoptosis (24), we demonstrated that shFAT1 significantly down-regulated the expression levels of marker of proliferation Ki-67, the proliferating cell nuclear antigen (PCNA) and B cell lymphoma 2 (Bcl2) proteins, while up-regulating Bax protein expression in the transfected SAS and HSC-3 cells, compared to their wild-type counterparts (Figure 4C), which is suggestive of a role for FAT1 in the promoting OSCC cell proliferation and evasion of cell death.

As suggested by the reviewer,we have included a brief mention of the rationale for measuring GSH in our revised manuscript. Please kindly see our revised Results section, Page 12, Lines 456-468:

Our results indicated that compared to the wild-type (WT) cells, the HSC-3 CispR and SAS CispR were significantly less responsive to cisplatin treatment, with concentration as high as 40 M eliciting only a ~38.9% (vs. 68.1% in WT, p < 0.05) or 41.5% (vs. 87.4% in WT, p < 0.01) reduction in the viability of the HSC-3 CispR and SAS CispR, respectively (Figure 6A). Thereafter, we demonstrated that FAT1 is implicated in this cisplatin resistance phenotype with a 7.4-fold higher mRNA expression in the HSC-3 CispR cells compared to the HSC-3 WT cells (p < 0.01) (Figure 6B), and a 3.7-fold (p < 0.01) or 4.8-fold (p < 0.01) upregulated FAT1 protein expression in the SAS CispR or HSC-3 CispR cells compared to their parental/wild-type counterparts (Figure 6C). Consistent with the data above, and in line with contemporary knowledge implicating increased glutathione (GSH) levels in tumor initiation, disease progression, increased metastasis, and the chemoresistant stem cell-like phenotype of cancerous cells (25-27), we showed that the median intracellular GSH level in the FAT1-rich HSC-3 CispR cells was 2.38-fold (p < 0.01) fold higher than in the HSC-3 wild-type cells (Figure 6D).

Please also kindly see our revised Results section, Page 17, Lines 635-659:

Contextually, our FAT1 findings are, at least in part, corroborated by data indicating that the aberrant expression of the male-specific protocadherin-PC (PCDH-PC) in prostate cancer cells facilitate their acquisition of an apoptosis-evading and hormone therapy-resistant phenotype through enhanced nuclear accumulation of b-catenin and increased WNT - signaling (40). Also, the observed strong co-expression of FAT1, molecular components of the LRP5 signalosome, namely LRP8, GSK3b, WNT2, b-catenin, casein kinase 1 gamma 1 (CSNK1G1), CSNK1G2, CSNK1G3, AXIN1, caveolin (CAV)1, CAV2, and glutathione (GSS), with weak expression of glutathione-disulfide reductase (GSR) in patients with OSCC, compared with the dysplasia or normal control group, is partially in concordant with reports demonstrating a positive correlation between expression of CAV1 and LRP5-analogous LRP6 in human primary and metastatic prostate cancer tissues, and that the interaction between CAV1 and LRP6 plays an important role in the regulation of Wnt/β-catenin signaling (41).

Moreover, our finding demonstrating that the ablative targeting of FAT1 re-sensitizes cisplatin-resistant OSCC cells to cisplatin through suppressed glutathione (GSH) expression, enhanced oxidative stress and deregulation of LRP/Wnt signaling, become therapeutically relevant when put in the context of contemporary knowledge that while resistance to chemotherapeutics constitute a major impediment to treatment success, cisplatin-resistance can be overcome by inhibition of glutathione S-transferase (GSTs), in vitro and in vivo, where the activity of GST is dependent on the steady production or availability of glutathione (GSH) (25, 42, 43), and deregulated GSH homeostasis is implicated in the pathogenesis and progression of several human diseases including malignancies, especially as impaired GSH production, or decreased GSH/glutathione disulphide (GSSG) ratio, results in enhanced susceptibility to oxidative stress, which in turn is culpable in cancer progression, where elevated GSH levels augment the antioxidant capacity and resistance to oxidative stress characteristic of many cancerous cells, including OSCC (25).

Q4.4.    The authors showed numerous factors that may be correlated with FAT1, it would be better to describe the reason to select the LRP/WNT signaling for further examination.

A4.4.    We appreciate the reviewer’s comment. As rightly noted by the reviewer, the present proof-of-principle study expounds the role of FAT1 in the modulation of the oral carcinogenesis, oxidative stress and cisplatin resistance via deregulation of the LRP5/WNT2/GSS signaling axis. All molecular factors alluded to in the study are components of the crosstalk between WNT/b-catenin and Glutathione oxidative stress signaling pathways. Please kindly see our revised Discussion section, Pages 17-18, Lines 643-686:

Contextually, our FAT1 findings are, at least in part, corroborated by data indicating that the aberrant expression of the male-specific protocadherin-PC (PCDH-PC) in prostate cancer cells facilitate their acquisition of an apoptosis-evading and hormone therapy-resistant phenotype through enhanced nuclear accumulation of b-catenin and increased WNT - signaling (41). Also, the observed strong co-expression of FAT1, molecular components of the LRP5 signalosome, namely LRP8, GSK3b, WNT2, b-catenin, casein kinase 1 gamma 1 (CSNK1G1), CSNK1G2, CSNK1G3, AXIN1, caveolin (CAV)1, CAV2, and glutathione (GSS), with weak expression of glutathione-disulfide reductase (GSR) in patients with OSCC, compared with the dysplasia or normal control group, is partially in concordant with reports demonstrating a positive correlation between expression of CAV1 and LRP5-analogous LRP6 in human primary and metastatic prostate cancer tissues, and that the interaction between CAV1 and LRP6 plays an important role in the regulation of Wnt/β-catenin signaling (42).

Moreover, our finding demonstrating that the ablative targeting of FAT1 re-sensitizes cisplatin-resistant OSCC cells to cisplatin through suppressed glutathione (GSH) expression, enhanced oxidative stress and deregulation of LRP/Wnt signaling, become therapeutically relevant when put in the context of contemporary knowledge that while resistance to chemotherapeutics constitute a major impediment to treatment success, cisplatin-resistance can be overcome by inhibition of glutathione S-transferase (GSTs), in vitro and in vivo, where the activity of GST is dependent on the steady production or availability of glutathione (GSH) (26, 43, 44), and deregulated GSH homeostasis is implicated in the pathogenesis and progression of several human diseases including malignancies, especially as impaired GSH production, or decreased GSH/glutathione disulphide (GSSG) ratio, results in enhanced susceptibility to oxidative stress, which in turn is culpable in cancer progression, where elevated GSH levels augment the antioxidant capacity and resistance to oxidative stress characteristic of many cancerous cells, including OSCC (26). Thus, consistent with Matés JM, et al’s exposition on the implication of oxidative stress for cell proliferation, apoptosis and carcinogenesis (45), the present study expounds the role of FAT1 in the modulation of oral carcinogenesis, oxidative stress and cisplatin resistance via deregulation of the LRP5/WNT2/GSS signaling axis, wherein all molecular factors alluded to in the study are molecular effectors of the crosstalk between WNT/b-catenin and GSH oxidative stress signaling pathways.

Put together, as summarized in the Graphical abstract, this present study uncovers a new role for FAT1 in OSCC oncogenesis and cisplatin resistance, with some mechanistic insight into this oncogenic role, thereby highlighting the functional duality of FAT1 in OSCC. Our findings provide pre-clinical evidence for the therapeutic exploitation of FAT1 ambivalence especially in the treatment of metastatic and/or cisplatin-resistant disease.

We sincerely wish to thank the reviewer for taking time off his busy schedule to review our work and provide several constructive critiques and valuable suggestions. We have made use of all these suggestions and believe that they have helped improve the scientific quality and appeal of our work. We are indeed grateful.

Round 2

Reviewer 1 Report

Authors respond to my questions well. I have no further questions.

Reviewer 3 Report

While authors have addressed some minor comments, the main short coming (i.e. very low number of samples) has not been improved.